# Conformational equilibria and intrinsic affinities define integrin activation

Jing Li[1,2,†], Yang Su[1,2,†], Wei Xia[1,2], Yan Qin[1,2], Martin J Humphries[3], Dietmar Vestweber[4], Carlos Cabañas[5] (ID) & Chafen Lu[1,2] & Timothy A Springer[1,2,*] (ID)

## Abstract

We show that the three conformational states of integrin $\alpha_5\beta_1$ have discrete free energies and define activation by measuring intrinsic affinities for ligand of each state and the equilibria linking them. The 5,000-fold higher affinity of the extended-open state than the bent-closed and extended-closed states demonstrates profound regulation of affinity. Free energy requirements for activation are defined with protein fragments and intact $\alpha_5\beta_1$. On the surface of K562 cells, $\alpha_5\beta_1$ is 99.8% bent-closed. Stabilization of the bent conformation by integrin transmembrane and cytoplasmic domains must be overcome by cellular energy input to stabilize extension. Following extension, headpiece opening is energetically favored. N-glycans and leg domains in each subunit that connect the ligand-binding head to the membrane repel or crowd one another and regulate conformational equilibria in favor of headpiece opening. The results suggest new principles for regulating signaling in the large class of receptors built from extracellular domains in tandem with single-span transmembrane domains.

**Keywords** affinity; conformation; integrin; N-glycan; thermodynamics
**Subject Categories** Cell Adhesion, Polarity & Cytoskeleton; Structural Biology
**The EMBO Journal (2017) 36: 629–645**

## Introduction

To quantitatively relate the steps involved in signal transmission across the plasma membrane in cell surface receptors, an understanding of receptor energy landscapes is essential. The gaps in free energy between signal-competent and incompetent receptor conformational states are especially important. However, such understanding is currently limited to receptors with large lipid-embedded domains, such as G protein-coupled receptors and ion channels (Ruiz & Karpen, 1997; Horrigan et al, 1999; Lape et al, 2008; Park et al, 2008; Cecchini & Changeux, 2015; Manglik et al, 2015). Much less is known about receptors that bind ligands through extracellular domains that are in tandem with single-span transmembrane domains, including integrins. Integrins are cell surface receptors that mediate cell-to-cell and cell-to-matrix adhesion. Integrins contain $\alpha$-subunits and $\beta$-subunits that non-covalently associate into $\alpha\beta$ heterodimers (Springer & Dustin, 2012) (Fig 1A). The $\beta$-propeller domain in $\alpha$ and the $\beta$I domain in $\beta$ associate to form a ligand-binding head, which is linked to leg, transmembrane, and cytoplasmic domains in each subunit.

Integrins undergo large-scale conformational changes (Springer & Dustin, 2012). In the bent-closed (BC) conformation, the integrin ectodomain folds at knees in the $\alpha$- and $\beta$-subunits so that the head and upper legs associate with the lower legs (Fig 1A). In two extended states, the extended-closed (EC) and extended-open (EO) conformations, extension of the $\alpha$- and $\beta$-knees raises the headpiece above the lower legs on cell surfaces (Fig 1A). In transition from EC to EO, that is, headpiece opening, the ligand-binding metal ion-dependent adhesion site (MIDAS) in the $\beta$-subunit $\beta$I domain rearranges. This reshaping of the ligand-binding site is linked by $\alpha$-helix pistoning within the $\beta$I domain to swing of the hybrid domain away from the integrin $\alpha$-subunit (Fig 1A). Although the affinities of these states have not yet been measured, previous studies have correlated integrin adhesiveness and high affinity for ligand with the EO conformation (Takagi et al, 2002, 2003; Xiao et al, 2004; Chen et al, 2010; Schürpf & Springer, 2011; Zhu et al, 2013). Measurements of affinities on receptors with multiple conformational states yield an average affinity that is weighted according to the population of each conformation in the ensemble (Fig 1C). To understand biological function and its regulation, we need to know the affinity of specific receptor conformational states (intrinsic affinities, $K_a$), and the equilibrium constants ($K_{conf}$) linking inactive conformation(s) to active conformation(s). These quantities, to the best of our knowledge, remain undetermined for integrins and the large class of receptors with single-span transmembrane domains. In the absence of

1 Program in Cellular and Molecular Medicine, Boston Children's Hospital, Boston, MA, USA
2 Department of Biological Chemistry and Molecular Pharmacology, Harvard Medical School, Boston, MA, USA
3 Wellcome Trust Centre for Cell-Matrix Research, University of Manchester, Manchester, UK
4 Max-Planck-Institute of Molecular Biomedicine, Münster, Germany
5 Centro de Biología Molecular Severo Ochoa (CSIC-UAM), and Departamento de Microbiología I, Facultad de Medicina, UCM, Madrid, Spain
*Corresponding author. Tel: +1 617 713 8200; E-mail: timothy.springer@childrens.harvard.edu
†These authors contributed equally to this work

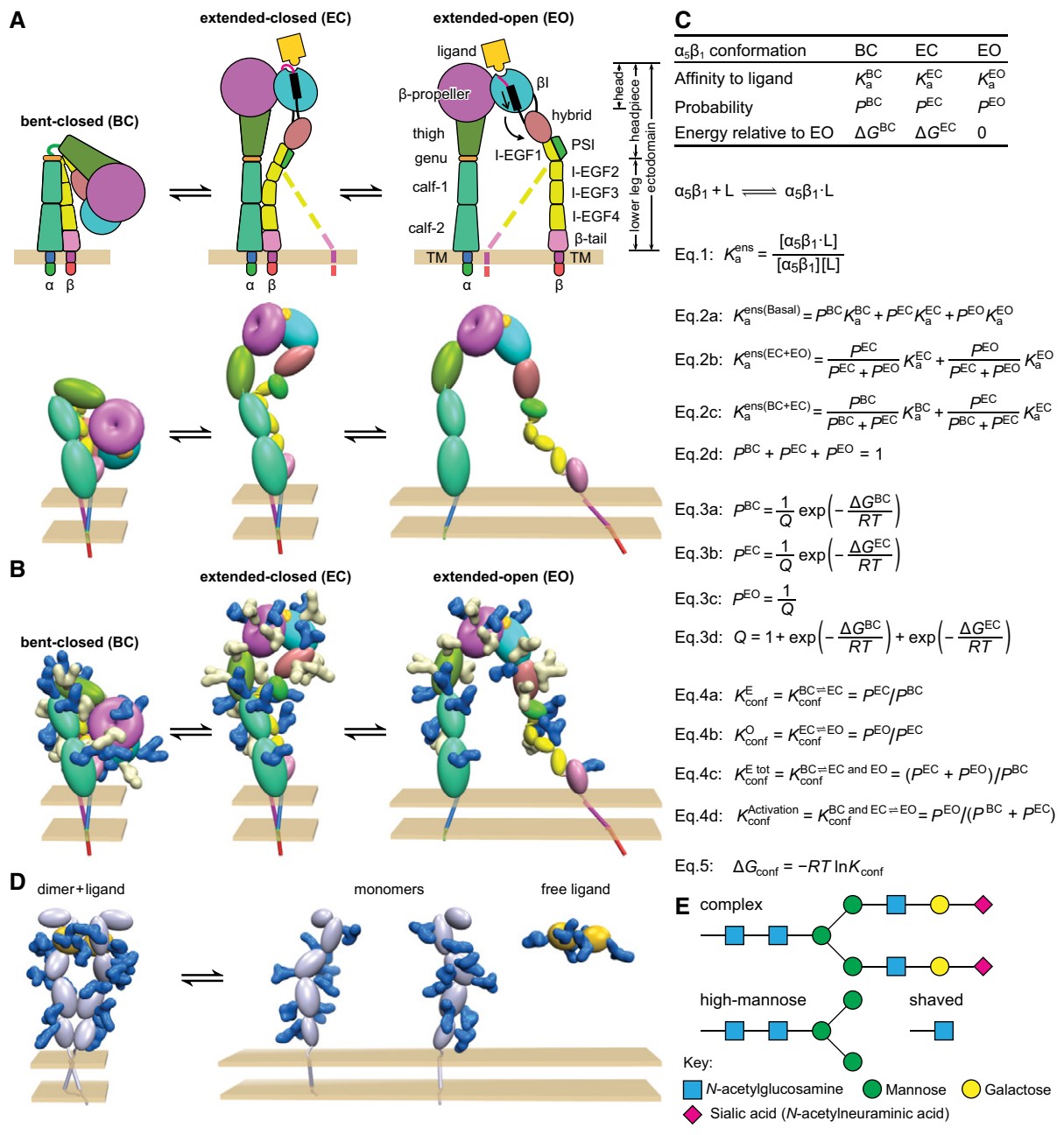

**Figure 1. Thermodynamic equilibria of integrin $\alpha_5\beta_1$ and relation to other receptors.**

A–C   Integrins. (A) Integrin structure and arrangement of domains in the three overall states of the conformational ensemble (Luo *et al*, 2007), with dashed lines representing the flexibility of the lower β-leg in extended conformations (schematic, upper). In the Pymol representation (lower), structures based on intact, bent-closed integrin $\alpha_V\beta_3$ on the cell surface (Zhu *et al*, 2009), in the same integrin subfamily as $\alpha_5\beta_1$, are shown using an ellipsoid or torus for each extracellular domain. The extended-closed structure is made by rigid body movements at the knees. The extended-open structure is derived from extended-closed by superposition on the open headpiece of $\alpha_{IIb}\beta_3$ (Xiao *et al*, 2004). (B) Identical to (A, lower) except molecular surfaces are shown for N-glycans removed by mutation (Fig 6D–F, white) or not removed (blue). Glycans have the structure shown in (E). Glycans were added at all $\alpha_5\beta_1$ N-glycosylation sequons using $\alpha_5\beta_1$ headpiece structures (Xia & Springer, 2014) and an $\alpha_V\beta_3$ homology model. (C) Equations that describe the affinities and conformational equilibria of integrin ensembles.

D   Representative cytokine or growth factor receptor. The ligand-bound dimer is compact similarly to the bent-closed integrin conformation, whereas the unassociated monomers show no interaction with one another and thus separate from one another similarly to leg domains in the extended-open integrin structure. Therefore, inter-subunit crowding interactions may regulate ligand binding and dimerization in such receptors analogously to regulation of conformational transitions in integrins. Individual domains are shown as ellipses, and N-glycans are shown as blue molecular surfaces using the same chemical structures as in (B).

E   Representative N-glycan structures. The complex glycan is common among $\alpha_5\beta_1$ N-glycans (Sieber *et al*, 2007) and contains the average number of monosaccharide residues per N-glycosylation sequon found here for the complex glycoform of $\alpha_5\beta_1$. The high-mannose glycan contains the average number of monosaccharide residues in the high-mannose form of $\alpha_5\beta_1$. The shaved glycan is that after Endo H digestion; the mass estimated here for shaved $\alpha_5\beta_1$ suggests part of high-mannose glycans are shaved and the remainder are resistant to Endo H.

measurement of the intrinsic affinities of specific integrin conformational states, the magnitude of the affinity increase is unknown (Zhu *et al*, 2013; Su *et al*, 2016). In the absence of knowledge of $K_{conf}$, we do not know which conformations predominate biologically, and how much energy is required to stabilize the active, high affinity state of integrins.

By measuring conformational equilibria here, we have also discovered that previously poorly appreciated components of surface receptors, such as their N-glycans and their leg domains that connect ligand-binding domains to the cell surface, can have important regulatory functions. Integrins, like most other cell surface receptors, are heavily glycosylated (Fig 1B). Glycans serve a variety of structural and functional roles (Stowell *et al*, 2015), but a compelling function is lacking in most receptors. The significance of the great variation in number of N-linked glycosylation sites among integrin subunits, ranging from 5 to 26 sites among the 18 human α-subunits and from 5 to 12 sites among the eight human β-subunits, is currently unknown. Moreover, during cell differentiation or transition from stasis to proliferation, alterations in glycosyl transferases cause substantial changes in N- and O-linked glycan processing, that is, in glycan branching and the number and nature of the monosaccharide residues in the glycan. Nonetheless, whether N-glycosylation regulates signaling by altering the equilibria between inactive and active receptor conformational states has been unknown.

Activation models in the integrin field have previously been discussed conceptually but not quantitatively. All integrin β-subunit cytoplasmic domains contain motifs that associate with cytoskeletal proteins. The β$_1$-subunit, among six of the eight mammalian integrin β-subunits, has binding sites for talins and kindlins, which link to the actin cytoskeleton (Calderwood *et al*, 2013). By an incompletely characterized process termed inside-out signaling, coupling through talin and kindlin to the actin cytoskeleton stabilizes the high affinity state of integrins (Springer & Dustin, 2012). In contrast, other proteins bind to integrin cytoplasmic domains and stabilize them in the inactive state (Bouvard *et al*, 2013). To quantitatively relate the steps involved in signal transmission across the plasma membrane in surface receptors, an understanding of the energy landscapes of their conformational ensembles is essential. The energy landscape, which dictates the fractional population of signaling competent and incompetent states of receptors on cell surfaces (Fig 1C), provides fundamental information including how much cellular energy is needed for integrin activation.

We use in our studies the model integrin α$_5$β$_1$, a receptor for fibronectin that contributes to the assembly of fibronectin into fibrils (Schwarzbauer & DeSimone, 2011). Within fibronectin, α$_5$β$_1$ recognizes an Arg-Gly-Asp (RGD) motif in a flexible loop in Fn3 domain 10 and a synergy site in Fn3 domain 9. We use allosteric, conformation-specific antibodies to convert the three overall conformational states basally present in the α$_5$β$_1$ integrin ensemble into either one or two defined states. Measurements of the ligand-binding affinities of these ensembles enable calculation of intrinsic affinities and free energies of each state for both purified α$_5$β$_1$ fragments and intact α$_5$β$_1$ on cell surfaces. Antibodies that were originally selected to inhibit, activate, or report the activation status of the 12 different β$_1$ integrin αβ heterodimers are essential tools in this work (Byron *et al*, 2009; Su *et al*, 2016). Electron microscopy (EM) and functional comparisons among these antibodies have defined the α$_5$β$_1$ conformational state(s) that they stabilize (Su *et al*, 2016).

Definition here of the energy landscape for a receptor with tandem extracellular, single-span transmembrane, and cytoplasmic domains reveals many features not previously anticipated.

## Results

### Principles for saturable stabilization of defined conformational states

Here, we describe the fluorescence polarization (FP) assay used in much of this work, and the methods required to establish that the measurements reflect values for the desired conformational states. FP is a rigorous, highly reproducible method for measuring binding of small fluorescent ligands to larger partners based on slowed tumbling and increased FP of the ligand-partner complex (Rossi & Taylor, 2011). Using fixed concentrations of integrin and fluorescein isothiocyanate (FITC)-labeled cyclic RGD peptide (cRGD) (Koivunen *et al*, 1995; Xia & Springer, 2014) and titrating in Fabs, we measured the ability of Fabs to increase or decrease ligand binding by altering integrin affinity for cRGD. We used Fabs (Fig 2A) to exclude complications from cross-linking by IgGs. Fab titrations showed that plateau values of FP were reached, reflecting saturable population of the desired conformations in the ensemble, and enabled calculation of Fab EC$_{50}$ values (Fig 2B). The results were concordant with the effect of Fabs on α$_5$β$_1$ conformation visualized by EM and on cell adhesion (Su *et al*, 2016). 12G10, HUTS4, and TS2/16 Fabs to β$_1$ increased FP to similarly high plateau values of 0.21–0.22 (Fig 2B), correlating with their stabilization of the open conformation of the βI domain (TS2/16) or the open conformation of the entire headpiece (12G10 and HUTS4) (Su *et al*, 2016). 8E3, 9EG7, and N29 Fabs to β$_1$ and SNAKA51 Fab to α$_5$ increased FP to lower plateau values of 0.14–0.16, consistent with the finding that they stabilize the extended conformation of α$_5$β$_1$ and hence stabilize the EC and EO states (Su *et al*, 2016). In contrast, two inhibitory Fabs, SG/19 and mAb13, decreased FP from 0.12 to plateau values of 0.09 (Fig 2B), in agreement with findings that they stabilize the closed conformation of the headpiece and hence stabilize the BC and EC states (Su *et al*, 2016).

To saturably populate desired conformations, Fabs must be used well above their EC$_{50}$ values; furthermore, experimental design must take into account the principle that EC$_{50}$ values are dependent on the population of the states in the pre-existing ensemble. We illustrate this by comparing EC$_{50}$ values in Mg$^{2+}$ to those in the presence of the integrin activator Mn$^{2+}$ (Fig 2C). The higher FP basal values in Mn$^{2+}$ (0.22) than in Mg$^{2+}$ (0.12) suggested a higher proportion of the high affinity, EO state of α$_5$β$_1$ in the conformational ensemble. Consistent with this higher proportion of the EO state in Mn$^{2+}$, EC$_{50}$ values for the HUTS4 Fab specific for the EO state and 8E3 Fab specific for the EC and EO states were lower than in Mg$^{2+}$. Conversely, consistent with the lower proportion of the BC and EC states in Mn$^{2+}$, the EC$_{50}$ value for SG/19 Fab specific for the EC and BC states was higher than in Mg$^{2+}$ (Fig 2C). Accordingly, we have measured Fab EC$_{50}$ values for each type of α$_5$β$_1$ preparation studied in the experiments in this paper (Fig 2, and Appendix Fig S1 and Appendix Table S1). In all experiments, Fabs are used at concentrations well above their EC$_{50}$, such that the population of the state(s) they stabilize approaches 100% and apparent affinity

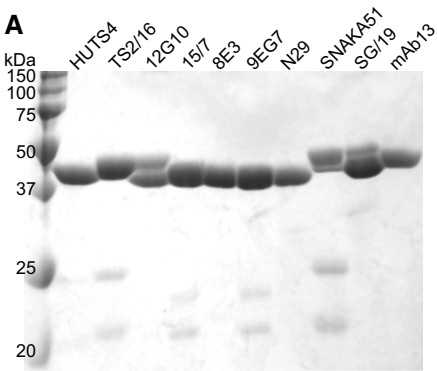

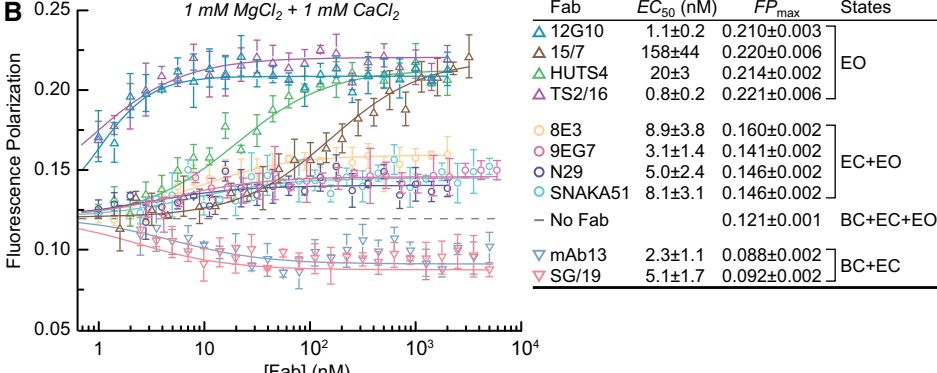

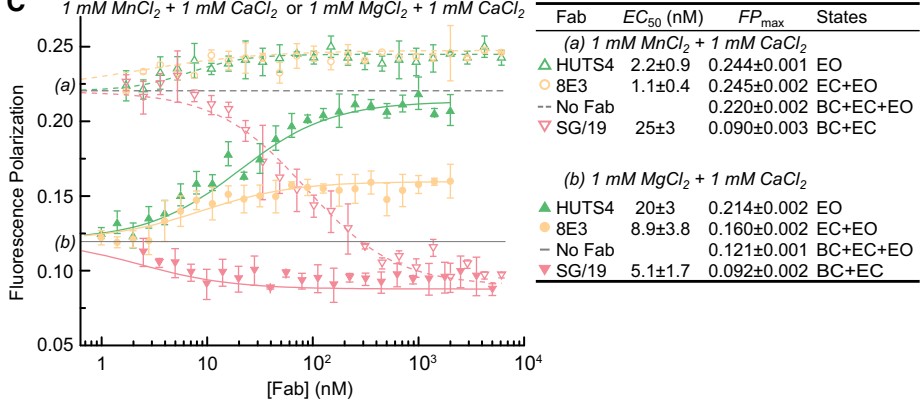

**Figure 2.  Influence of conformation-selective Fabs and metal ions on ligand binding by the α₅β₁ ectodomain.**

A    Coomassie-stained, non-reducing SDS 12.5% PAGE of Fabs.

B, C    Dependence of unclasped, high-mannose α₅β₁ ectodomain (20 nM) binding to FITC-cRGD (5 nM) in FP assays on Fab concentration in 1 mM Mg²⁺ & 1 mM Ca²⁺ (B) and in 1 mM Mg²⁺ & 1 mM Ca²⁺ compared to 1 mM Mn²⁺ & 1 mM Ca²⁺ (C). Values to the right of each plot are from nonlinear least square fits to Appendix Equation S1 (mean ± s.d. of triplicates).

$(K_d^{app})$ approaches true ensemble affinity $(K_d^{ens})$ (Appendix Table S1 and Appendix Fig S2).

A correction is required for measurements with Fabs that stabilize the closed conformations, because even a small amount of the much higher affinity open conformation makes a substantial contribution to measured apparent affinities. For these conditions, Fab $K_d$ values or Fab EC₅₀ values measured under conditions where they closely approximate $K_d$ are used to convert measured $K_d^{app}$ values to $K_d^{ens}$ values (Appendix Figs S1 and S2, Appendix Table S1 and Appendix Equations S47–S55).

Finally, we evaluated the assumption that the intrinsic affinities of Fab-bound states measured here are close to the intrinsic affinity of that state in the absence of Fab. We tested a corollary: if Fabs stabilize states distinct from those in the absence of Fab, then states stabilized by distinct Fabs should also differ in ligand-binding affinity from one another. We therefore began our studies with an integrin ectodomain preparation in which ensembles containing the closed (BC+EC), extended (EC+EO), and EO conformations were well separated in affinity, enabling us to test the assumption that Fabs that stabilize the same states should give similar integrin

ligand-binding affinities. The $\alpha_5\beta_1$ ectodomain was purified from cells that secrete glycoproteins with high-mannose N-glycans. A clasp between the C-termini of the $\alpha_5$ and $\beta_1$-subunits that facilitated high expression was proteolytically cleaved to yield the unclasped, high-mannose $\alpha_5\beta_1$ ectodomain. We used FP with a fixed concentration of FITC-cRGD in the absence or presence of Fabs at concentrations well above their $EC_{50}$ values and measured integrin ligand-binding affinity by titrating in the integrin. The affinity for cRGD of the basal $\alpha_5\beta_1$ ensemble in the absence of Fab, that is, $K_d^{ens(Basal)}$, the population-weighted average affinity of all three states as shown in Eq. 2a in Fig 1C, was 47 nM (Fig 3A). Note that for concision, we use association constant $K_a$ in equations and dissociation constant $K_d = 1/K_a$ for reporting affinities. The affinity intrinsic to the EO conformation was $K_d^{EO} = 2.0$–2.6 nM, measured in the presence of open-stabilizing Fabs TS2/16, 12G10, or HUTS4 (Fig 3A). The affinity intrinsic to the EC conformation was $K_d^{EC} = 6,500$–9,200 nM, measured in the presence of two sets of mutually compatible extension-stabilizing and closure-stabilizing Fabs (Su *et al*, 2016), SNAKA51 plus SG/19, or 9EG7 plus mAb13 (Fig 3A). The affinity of the ensemble comprising the two extended

conformations, EC and EO (Eq. 2b in Fig 1C), was $K_d^{ens(EC+EO)} = 15$–22 nM (Fig 3A), measured in the presence of extension-stabilizing Fabs 8E3, N29, 9EG7, or SNAKA51. The affinity of an ensemble comprising the two closed conformations, BC and EC (Eq. 2c in Fig 1C), was $K_d^{ens(BC+EC)} = 4,600$–9,400 nM, measured in the presence of closure-stabilizing Fab SG/19 or mAb13 (Fig 3A). The similar affinities measured with independent Fabs stabilizing the same state, compared to the distinct affinities measured for the EO state, the closed states, and the extended states, support the assumption that these Fab-stabilized states are similar to the native states.

Affinities for cRGD in the absence or presence of Fabs were independently measured with the $\alpha_5\beta_1$ headpiece, which lacks the lower legs of the ectodomain (Fig 1A). $K_d^{ens(Basal)}$ was almost 100-fold lower for the headpiece than the ectodomain (Fig 3A and B). Nonetheless, the intrinsic affinity of the open headpiece, $K_d^O = 1.9 \pm 0.3$ nM (Fig 3B), was very similar to that of the extended-open ectodomain, $K_d^{EO} = 2.0$–2.6 nM (Fig 3A). The affinity of the closed headpiece, $K_d^C = 8,500$–9,600 nM, with SG/19 and mAb13 Fabs (Fig 3B) was in agreement with the lower estimate of 9,400 nM for closed ectodomain conformations (Fig 3A). Because

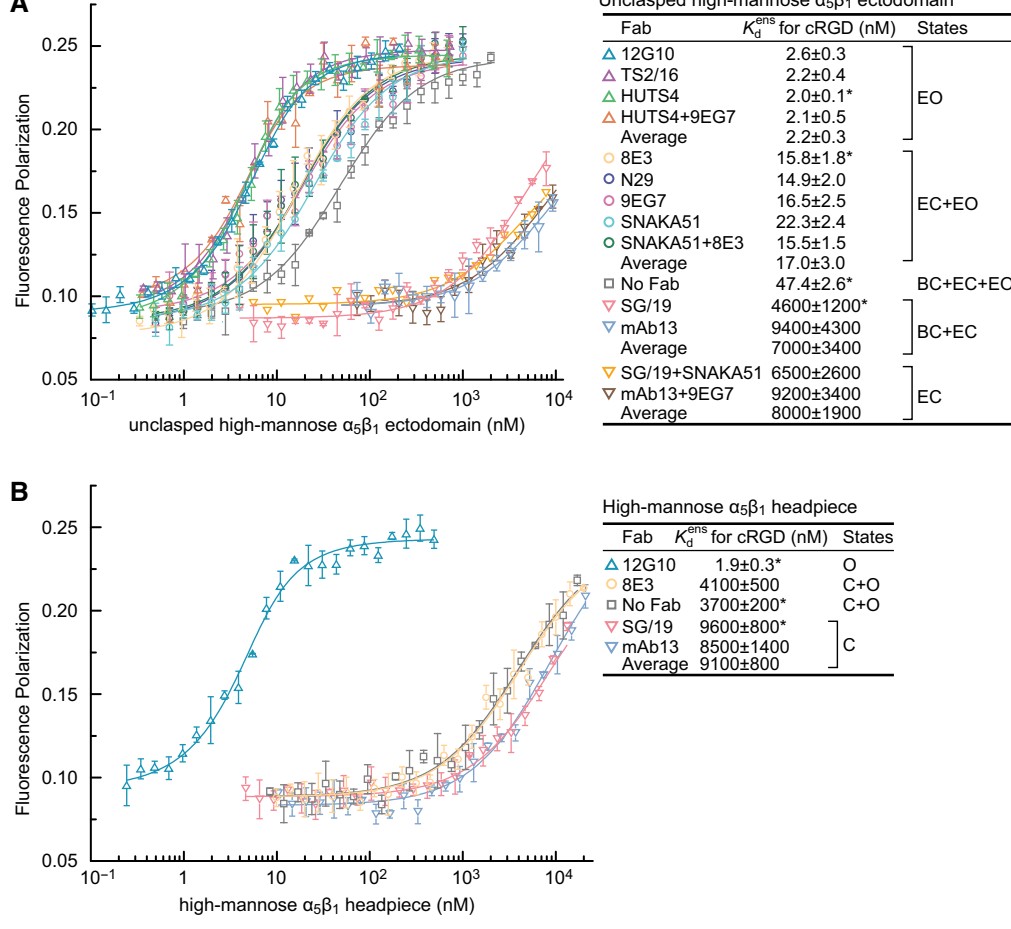

**Figure 3.  Intrinsic and ensemble affinities of $\alpha_5\beta_1$ ectodomain and headpiece preparations for cRGD.**

A, B  Affinities of high-mannose glycoforms of the unclasped $\alpha_5\beta_1$ ectodomain (A) and headpiece (B) were measured using FP with FITC-cRGD in the presence of the indicated Fabs. Errors in the plotted datapoints are s.d. from average value of triplicate measurements. Errors for affinities in the inset table are s.d. from nonlinear least square fits of the average values from the triplicate measurements except values with "*" are s.d. from three experiments on different days.

the lower α- and β-legs are absent in the headpiece, Fabs that bind to the upper β-leg and stabilize extension (Fig 1A) should have no effect on headpiece affinity for ligand. Indeed, binding of extension-stabilizing Fab 8E3 to the PSI domain resulted in no change in head-piece affinity for cRGD (Fig 3B) in contrast to the threefold increase in affinity with the ectodomain (Fig 3A), even though 8E3 binds well to the headpiece as shown with ITC (Appendix Fig S3A).

## Independent ligands and the free energy of each conformational state

Having established the validity of our use of conformation-specific Fabs to measure the properties of specific integrin conformational states, we went on to test further concepts and examine the molecular features that regulate integrin affinity and conformational equilibria. Distinct ligands are expected to have distinct affinities intrinsic to each integrin conformational state; however, the free energies of these states should be identical, if the ligands indeed bind to the same conformational states. We tested these ideas with two more ligands of $\alpha_5\beta_1$, the peptide GRGDSPK (RGD) and a fragment of fibronectin containing Fn3 domains 9 and 10 (Fn3$_{9-10}$). FP measurements with FITC-RGD showed $K_d^{EO}$ of 71 nM, $K_d^{ens(EC+EO)}$ of 620 nM, and $K_d^{ens(Basal)}$ of 2,300 nM (Fig 4A). ITC measurements with RGD peptide showed $K_d^{EO}$ of 68 nM and $K_d^{ens(Basal)}$ of 2,100 nM (Appendix Fig S3B and C). This excellent agreement in affinity between FP for FITC-RGD and ITC for RGD is more than sufficient to validate the energy landscape conclusions below. Fn3$_{9-10}$ affinity for $\alpha_5\beta_1$ was measured by competing FITC-cRGD binding in FP (Fig 4B and C). Fn3$_{9-10}$ showed $K_d^{EO}$ of 0.44 nM, $K_d^{ens(EC+EO)}$ of 2.0 nM, $K_d^{ens(Basal)}$ of 5.2 nM, $K_d^{ens(BC+EC)}$ of 2,900 nM, and $K_d^{EC}$ of 2,700 nM (Fig 4B–D). These measurements, and those in Fig 3, suggest that $K_d^{EC}$ and $K_d^{ens(BC+EC)}$ are indistinguishable from one another, and by extension, that $K_d^{EC}$ and $K_d^{BC}$ are indistinguishable.

The population of each conformational state in the basal ensemble was calculated from $K_d^{ens(Basal)}$, $K_d^{ens(BC+EC)}$, $K_d^{ens(EC+EO)}$, and $K_d^{EO}$ (Eq. 2a–d in Fig 1C) and was insensitive to the relatively large uncertainty in $K_d^{ens(BC+EC)}$ (Appendix Fig S2B). The population of each conformational state relates through the Boltzmann distribution to the relative free energy of each state and was thus used to calculate $\Delta G$ (Eq. 3a–d in Fig 1C). For each $\alpha_5\beta_1$ preparation, EO is used as the reference state ($\Delta G^{EO} = 0$). Despite use of three ligands varying more than 100-fold in affinity, saturation binding with cRGD and RGD, as well as competition with Fn3$_{9-10}$, yielded $\Delta G$ values that were within experimental error of one another, demonstrating the robustness of the results (Fig 4D and E). With all three ligands, the BC state was lowest in energy, and the EC state was intermediate in energy between the BC and EO states. The most accurately determined values, with cRGD, showed $\Delta G^{BC} = -1.5 \pm 0.1$ kcal/mol and $\Delta G^{EC} = -1.1 \pm 0.1$ kcal/mol with $\Delta G^{EO} = 0$ kcal/mol (Fig 4D and E).

Population of each conformational state also defined the conformational equilibria ($K_{conf}$) between the states (Eq. 4a–d in Fig 1C) and enabled calculation of the free energies associated with integrin conformational change ($\Delta G_{conf}$) (Eq. 5 in Fig 1C). Thus, the extension step from BC to EC costs 0.4 kcal/mol ($\Delta G_{conf}^E$) and the opening step from EC to EO costs 1.1 kcal/mol ($\Delta G_{conf}^O$) (Fig 4D). Interconversion among the three integrin states does not necessarily happen

in a defined order (Takagi et al, 2002; Sen et al, 2013) (Movies EV1–EV3) and thus may also be conceptualized as interchange between one state and two other states. Thus, we may also consider extension as occurring from BC to either EC or EO ($K_{conf}^{Etot}$ and $\Delta G_{conf}^{Etot}$, Eq. 4c in Fig 1C). $\Delta G_{conf}^{Etot}$ (0.3 kcal/mol) was similar to $\Delta G_{conf}^E$ (0.4 kcal/mol) for unclasped, high-mannose $\alpha_5\beta_1$ (Fig 4D). Similarly, we may define $K_{conf}^{Actiation}$ and $\Delta G_{conf}^{Activation}$ for conversion from either BC or EC to EO (Eq. 4d in Figs 1C and 4D).

## Regulation of conformational equilibria by C-terminal clasp and N-glycosylation

We next examined molecular features that regulate the populations (and relative free energies) of the three overall integrin conformational states. The C-termini of the integrin α- and β-subunit ectodomains are close to one another in bent ectodomain crystal structures and are followed in sequence by the α- and β-subunit transmembrane (TM) domains that associate with one another. Complementary α-helical sequences appended to the α and β ectodomains that associate as coiled-coils are frequently used to clasp the C-termini together in this region. The clasp can be released (unclasped) by protease digestion at a specific site included in the linker between the ectodomain and coiled-coils. EM of distinct integrin αβ heterodimers has shown that the clasp increases the proportion of particles in the bent conformation relative to extended conformations (Takagi et al, 2002; Nishida et al, 2006; Springer & Dustin, 2012).

We also tested whether N-glycans regulated integrin conformational equilibria by comparing $\alpha_5\beta_1$ with native, that is, complex N-glycans (Sieber et al, 2007), high-mannose N-glycans, and shaved N-glycans. Shaving with endoglycosidase H of integrins with high-mannose N-glycans leaves only a single monosaccharide residue attached to N-glycosylation sites, except for a minority of inaccessible sites (Xie et al, 2010). Testing clasped and unclasped $\alpha_5\beta_1$ ectodomains each with three types of N-glycans (Fig 5A), we obtained six cRGD binding datasets (Appendix Fig S4A–F) enabling intrinsic affinity and $\Delta G$ value determinations (Fig 5B and C) that are discussed as a whole.

The $K_d^{ens(Basal)}$ of the six preparations ranged from 12 to 200 nM, showing profound regulation of ensemble affinity by the clasp and N-glycans (Fig 5B). In contrast, the intrinsic affinities of the extended-open conformations in each preparation were very similar, with an average value of 2.4 nM (Fig 5B). Additionally, the affinities of the closed conformations, $K_d^{ens(BC+EC)}$, of each of the six preparations were similar to one another with an average value of 4,700 nM (Fig 5B). In agreement with the intrinsic affinities measured for headpiece and ectodomain constructs above, these results show that intrinsic affinities are properties of the ligand-binding site of specific integrin conformations and are influenced little, at least for peptide ligand and macromolecule fragments, by decorations at N-glycosylation sites or the distal clasp restraint.

The C-terminal clasp lowered $K_d^{ens(Basal)}$ of each integrin glycoform by several fold. The clasp markedly increased $\Delta G_{conf}^E$ by 1.1–1.5 kcal/mol (Fig 5B), showing that proximity of the α- and β-subunit C-termini as observed in bent ectodomain crystal structures indeed stabilizes the bent conformation. In contrast, the clasp had no consistent effect on the energy of headpiece opening ($\Delta G_{conf}^O$, Fig 5B).

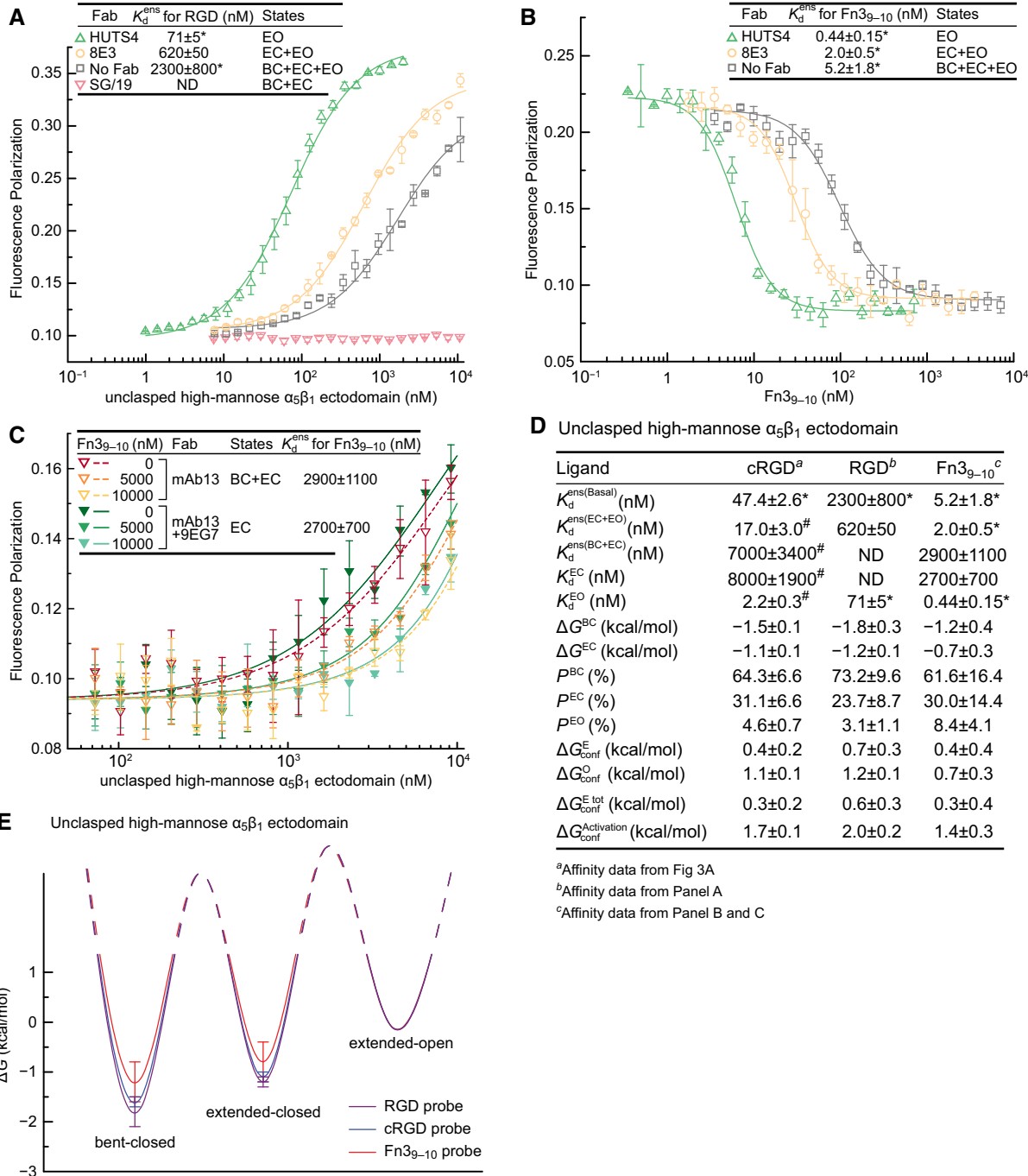

**Figure 4.  Intrinsic affinities and conformational equilibria of α5β1 with three different ligands.**

Data are for the unclasped, high-mannose α5β1 ectodomain using the indicated Fabs.

A    Affinity for FITC-RGD measured with FP.

B    Affinity for Fn3_9–10 measured by inhibition of FITC-cRGD binding to 20 nM α5β1 (with open-stabilizing Fab), 90 nM α5β1 (with extension-stabilizing Fab), or 270 nM α5β1 (with no Fab).

C    Affinity for Fn3_9–10 measured by inhibition of FITC-cRGD binding to a range of α5β1 concentrations (with closure-stabilizing Fab) at three Fn3_9–10 concentrations.

D    Tabulation of results.

E    Energy landscape plots showing valleys for the ΔG values determined here as lines, and hills for the transition state ΔG values, which remain to be determined, as dashed lines.

Data information: (A–E) Errors in the plotted datapoints are s.d. from the average value of triplicate measurements. Errors in the affinity values are s.d. from nonlinear least square fits of the average value of triplicate measurements except values with "*" are s.d. from three experiments on different days and values with "#" represent s.d. of measurements with distinct Fabs stabilizing the same conformation. Errors for *P* and ΔG values are propagated from affinities values as described in the Materials and Methods.

**A**

α₅β₁ ectodomain (non-reducing SDS–PAGE)

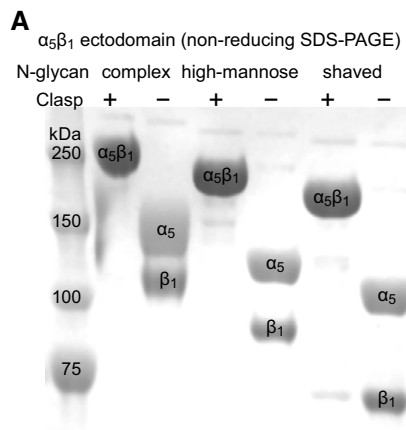

**B** α₅β₁ ectodomain

| Glycoform | Complex | | High-Mannose | | Shaved | |
|---|---|---|---|---|---|---|
| C-terminal Clasp | Clasped[a] | Unclasped[a] | Clasped[a] | Unclasped[b] | Clasped[a] | Unclasped[a] |
| $K_d^{ens(Basal)}$ (nM) | 24.0±3.5* | 11.7±3.7* | 151±55* | 47.4±2.6* | 202±60* | 109±16* |
| $K_d^{ens(EC+EO)}$ (nM) | 3.6±0.4* | 5.6±1.2* | 7.9±0.5* | 17.0±3.0# | 19.9±2.8* | 45±7* |
| $K_d^{ens(BC+EC)}$ (nM) | 5700±3100 | 3300±1100 | 4700±2000 | 7000±3400# | 5000±1200 | 4900±1100 |
| $K_d^{EO}$ (nM) | 2.6±0.7* | 3.6±0.3* | 2.3±0.6* | 2.2±0.3# | 1.9±0.3* | 2.2±0.3* |
| $\Delta G^{BC}$ (kcal/mol) | −1.2±0.2 | −0.3±0.4 | −2.4±0.3 | −1.5±0.1 | −2.7±0.2 | −2.0±0.2 |
| $\Delta G^{EC}$ (kcal/mol) | 0.6±0.6 | 0.3±0.4 | −0.5±0.2 | −1.1±0.1 | −1.3±0.1 | −1.7±0.1 |
| $P^{BC}$ (%) | 85.1±2.7 | 52.2±18.3 | 94.9±1.9 | 64.3±6.6 | 90.5±3.2 | 59.3±8.9 |
| $P^{EC}$ (%) | 4.2±3.4 | 17.1±11.9 | 3.6±1.4 | 31.1±6.6 | 8.6±3.0 | 38.8±8.7 |
| $P^{EO}$ (%) | 10.8±3.3 | 30.7±10.1 | 1.5±0.7 | 4.6±0.7 | 0.9±0.3 | 2.0±0.4 |
| $\Delta G_{conf}^{E}$ (kcal/mol) | 1.8±0.5 | 0.7±0.6 | 1.9±0.2 | 0.4±0.2 | 1.4±0.2 | 0.2±0.2 |
| $\Delta G_{conf}^{O}$ (kcal/mol) | -0.6±0.6 | −0.3±0.4 | 0.5±0.2 | 1.1±0.1 | 1.3±0.1 | 1.7±0.1 |
| $\Delta G_{conf}^{E\,tot}$ (kcal/mol) | 1.0±0.1 | 0.1±0.4 | 1.7±0.2 | 0.3±0.2 | 1.3±0.3 | 0.2±0.2 |
| $\Delta G_{conf}^{Activation}$ (kcal/mol) | 1.2±0.2 | 0.5±0.3 | 2.4±0.3 | 1.7±0.1 | 2.7±0.2 | 2.3±0.1 |

[a]Affinity data from Appendix Fig S4
[b]Affinity data from Fig 3

**C**

α₅β₁ ectodomain
— clasped, — unclasped shaved
— clasped, — unclasped high-mannose
— clasped, — unclasped complex

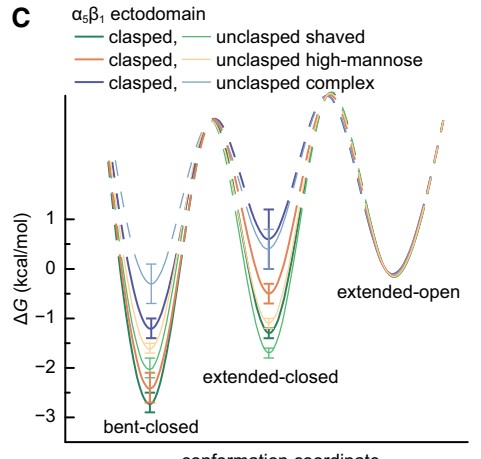

**D** Clasped complex N-glycan α₅β₁ ectodomain

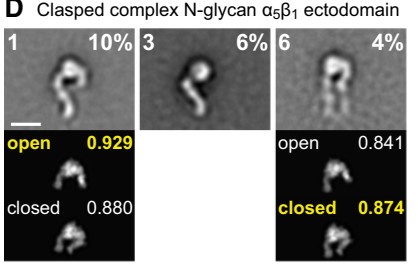

**E** Clasped shaved N-glycan α₅β₁ ectodomain

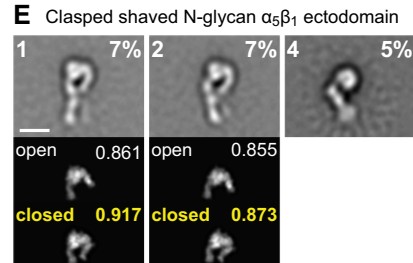

**Figure 5.  Glycoform and C-terminal clasp regulate α₅β₁ ectodomain conformational equilibria and have little effect on intrinsic affinities.**

A    SDS 7.5% PAGE of three α₅β₁ ectodomain glycoforms with or without C-terminal clasp stained with Coomassie blue.

B, C    Tabulation of results (B) and energy landscape plots (C). See Appendix Fig S4 for representative FP results. Error definitions and explanations of values marked with "*" and "#" are as in the Fig 4 legend.

D, E    Effect of complex (D) and shaved (E) N-glycans on clasped α₅β₁ ectodomain conformation in EM. Representative, well-resolved class averages are shown together with their rank among 35 class averages and the % of total particles in that class (inset, upper left and right, respectively). Below each *en face* class average are the best cross-correlating projections of open and closed headpiece crystal structures to the headpiece portion of the EM class average. Insets show correlation coefficients with the best correlating conformation in yellow. Scale bars are 10 nm. All class averages are shown in Appendix Fig S6.

N-linked glycoform type has a large influence on integrin α₅ and β₁-subunit migration in SDS–PAGE (Fig 5A). Multi-angle light scattering showed that complex, high-mannose, and shaved α₅β₁ ectodomain glycoprotein preparations are 21, 11, and 3% carbohydrate by mass, respectively (Appendix Fig S5). Moreover, N-glycoform has a surprisingly strong influence on conformational equilibria; free energies of the bent-closed and extended-closed conformations increase markedly with increasing % carbohydrate (Fig 5B). Compared to shaved N-glycans, complex N-glycans raised $\Delta G^{BC}$ by 1.5 kcal/mol for the clasped ectodomain and 1.7 kcal/mol for the unclasped ectodomain. Complex N-glycans also raised $\Delta G^{EC}$ by 1.9 kcal/mol (clasped) or 2.0 kcal/mol (unclasped) (Fig 5B). Remarkably, complex glycans reverse the relative stabilities of the two extended states so that the extended-open conformation is more stable than the extended-closed conformation (Fig 5C).

In negative stain EM, the high-mannose, clasped α₅β₁ ectodomain shows bent and extended-closed conformations (Su *et al*, 2016). Our findings on glycoforms led us to also examine the conformations of clasped α₅β₁ ectodomains with shaved and complex N-glycans (Fig 5D and E, and Appendix Fig S6). Both glycoforms exhibited bent and extended-closed α₅β₁ conformations; however, only α₅β₁ with complex N-glycans exhibited the extended-open conformation (Appendix Fig S5). Furthermore, the most populous class average of the clasped α₅β₁ ectodomain with complex N-glycans, representing 10% of all particles,

showed the extended-open conformation (Fig 5D panel 1). These EM results strongly support thermodynamic findings here that glycoforms regulate conformational equilibria and that complex N-glycans stabilize the EO state relative to the BC and EC states.

### Regulation of conformational equilibria by integrin lower legs and the number of N-linked glycosylation sites

Why would the presence of the lower legs in the integrin ectodomain raise ensemble affinity for ligand by stabilizing the open headpiece as shown above in Fig 3? We tested the hypothesis that crowding or repulsive interactions between the lower legs would favor the greater separation between the knees that is present in the open compared to the closed conformation (Fig 1A and B). Integrin $\alpha_5\beta_1$ containing the lower leg of the $\beta_1$-subunit and not that of the $\alpha_5$-subunit was well expressed and behaved during purification (Fig 6A). The $K_d^{ens(Basal)}$ of this semi-truncated integrin was identical to that of the headpiece, much lower than that of the ectodomain, and not affected by extension-stabilizing Fab (Fig 6B and C). Moreover, headpiece opening required 3.6 kcal/mol more energy for the semi-truncated ectodomain and headpiece constructs than for the unclasped ectodomain (Fig 6C).

The favoring of headpiece opening by increasing mass of N-glycans (Fig 5) might also suggest a role for crowding or repulsive interactions by glycans within the ectodomain. To test the corollary that the number of N-linked sites would similarly regulate headpiece opening, we mutated individual N-linked glycosylation sites in the integrin $\alpha_5$ and $\beta_1$-subunits. N-linked site mutations were tested individually, and those with the least effect on $\alpha_5\beta_1$ expression were combined to create $\Delta N\text{-}\alpha_5$ with 6 of 14 N-linked sites removed and $\Delta N\text{-}\beta_1$ with 5 of 12 sites removed. We then selected cell lines secreting all three possible combinations of mutant subunits with complex glycosylation (Fig 6D). $K_d^{ens(Basal)}$ of the $\Delta N\text{-}\alpha_5/\Delta N\text{-}\beta_1$ mutant was decreased threefold, whereas the $\Delta N\text{-}\alpha_5/\beta_1$ and $\alpha_5/\Delta N\text{-}\beta_1$ heterodimers showed intermediate decreases in affinity, as predicted by the crowding/repulsion hypothesis (Fig 6E and Appendix Fig S4G–I). Nonetheless, the intrinsic affinities of the extended-open conformation $K_d^{EO}$ stayed unchanged (Fig 6E and Appendix Fig S4). The bent-closed and extended-closed states were each stabilized by 0.8 kcal/mol in $\Delta N\text{-}\alpha_5/\Delta N\text{-}\beta_1$ relative to wild-type $\alpha_5\beta_1$ (Fig 6E and F).

### Conformational equilibria of cell surface $\alpha_5\beta_1$

We extended our measurements of affinities and conformational equilibria to native $\alpha_5\beta_1$ on the erythroleukemic cell line K562,

which expresses 200,000 $\alpha_5\beta_1$ molecules/cell (Faull *et al*, 1993), minimal levels of other integrin $\alpha$-subunits (Appendix Fig S7), and is completely dependent on integrin $\alpha_5\beta_1$ for adhesion to fibronectin (Tsuchida *et al*, 1997). Saturation binding of Alexa488-labeled Fn3$_{9-10}$ to K562 cells was measured by fluorescence flow cytometry with no washing (Chigaev *et al*, 2001; Dong *et al*, 2014). Extension-stabilizing and open-stabilizing Fabs greatly increased Fn3$_{9-10}$ binding to K562 cells (Fig 7A). $K_d^{ens(EC+EO)}$ was 1.9–2.1 nM, and $K_d^{EO}$ was 1.3–1.4 nM (Fig 7B), not far from values measured for the $\alpha_5\beta_1$ ectodomain in solution (Fig 4B). $K_d^{ens(Basal)}$ of K562 cells for Fn3$_{9-10}$ was too low to be measured by Alexa488-Fn3$_{9-10}$ saturation binding (Fig 7B). Therefore, we measured binding of Fn3$_{9-10}$ to K562 cells by enhancement of binding of open-stabilizing Alexa488-12G10 Fab, which yielded $K_d^{ens(Basal)}$ of 1,100 nM (Fig 7C).

Jurkat T lymphoblastoid cells express about eightfold less $\alpha_5\beta_1$ than K562 cells and higher levels of other $\beta_1$ integrins (Appendix Fig S8). Using the same methods as for K562 cells, we measured $\alpha_5\beta_1$-dependent binding affinities of Fn3$_{9-10}$ for Jurkat cells as $K_d^{ens(Basal)} = 750 \pm 60$ nM, $K_d^{ens(EC+EO)} = 1.7 \pm 0.2$ nM, and $K_d^{ens(EO)} = 1.6 \pm 0.2$ nM (Appendix Fig S8 and Fig 7E).

As a third, independent means of measuring the energy landscape of $\alpha_5\beta_1$ conformational states on cell surfaces, we measured the affinity of Alexa488-12G10 Fab for K562 cells. Use of 9EG7 Fab plus Fn3$_{9-10}$ to stabilize the extended-open conformation, or 9EG7 Fab plus SNAKA51 Fab to stabilize the ensemble with the two extended conformations, gave 12G10 Fab affinities of 0.4 and 0.7 nM, respectively (Fig 7D). In strong contrast, the affinity of 12G10 Fab for the basal ensemble of three states was 205 nM (Fig 7D).

The datasets on Fn3$_{9-10}$ and 12G10 Fab binding to K562 cells yield free energies of $\alpha_5\beta_1$ conformational states that are within experimental error of one another and similar to those on Jurkat cells (Fig 7E and F). On the cell surface, the BC state has a low free energy. $\Delta G^{BC}$ is $-4.0$ to $-3.6$ kcal/mol (Fig 7E). Thus, 99.7–99.8% of cell surface $\alpha_5\beta_1$ is in the bent-closed conformation. In contrast, the EC conformation is slightly higher in energy than the EO conformation, by 0.3–0.5 kcal/mol on K562 cells where it was much more accurately determined than on Jurkat cells. Thus, once $\alpha_5\beta_1$ extends on cell surfaces, it will predominantly populate the extended-open conformation, as found for the $\alpha_5\beta_1$ ectodomain with complex N-glycans with or without a clasp (Fig 5B). However, the bent-closed conformation is much more stable on cell surfaces than in clasped ectodomains (Fig 7F). Thus, the transmembrane and cytoplasmic domains of $\alpha_5\beta_1$, in the context of the plasma membrane and cytoplasmic environment, stabilize the bent-closed conformation much more than a coiled-coil clasp (Fig 7F and G).

---

**Figure 6.  Regulation of $\alpha_5\beta_1$ conformational equilibria by the lower legs and N-glycans.**

A–C  The semi-truncated $\alpha_5\beta_1$ ectodomain. (A) Coomassie-stained SDS 7.5% PAGE. (B) Affinity of FITC-cRGD for the semi-truncated, high-mannose $\alpha_5\beta_1$ ectodomain in the presence of the indicated Fabs measured with FP. (C) Comparison of the semi-truncated ectodomain to the unclasped $\alpha_5\beta_1$ ectodomain and headpiece, all with high-mannose glycans.

D–F  Effect of N-glycosylation sequon mutation. The $\Delta N\text{-}\alpha_5$ and $\Delta N\text{-}\beta_1$ mutants have 6 of 14 and 5 of 12 predicted N-glycosylation sites mutated, respectively. The indicated mutants were tested in the clasped $\alpha_5\beta_1$ ectodomain with complex N-glycosylation. (D) Coomassie-stained SDS 7.5% PAGE. (E) Tabulation of results. See Appendix Fig S4 for representative FP results. ND, not determined; limited solubility of the N-glycosylation sequon mutants prevented their use at the high concentrations required for intrinsic affinity measurements of the BC+EC ensemble. Therefore, thermodynamic parameters were calculated by assuming that the intrinsic affinities of the BC and EC conformations were identical to those determined for the WT ectodomain. (F) Energy landscape plots are as described in the Fig 4E legend.

Data information: (B, C, E, F) Error definitions are as described in the legend to Fig 4.

---

**A** Unclasped high-mannose $\alpha_5\beta_1$

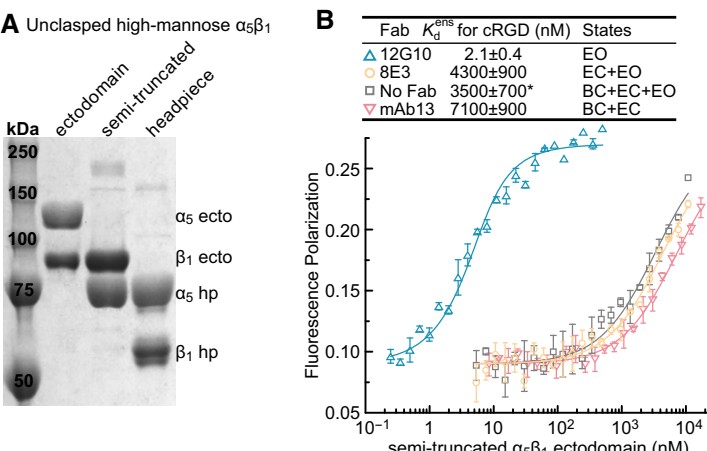

**B**

| Fab | $K_d^{ens}$ for cRGD (nM) | States |
|-----|-----|-----|
| △ 12G10 | 2.1±0.4 | EO |
| ○ 8E3 | 4300±900 | EC+EO |
| □ No Fab | 3500±700* | BC+EC+EO |
| ▽ mAb13 | 7100±900 | BC+EC |

**C** Unclasped high-mannose $\alpha_5\beta_1$

| | Ectodomain[a] | Semi-truncated[b] | Headpiece[c] |
|---|---|---|---|
| $K_d^{ens(Basal)}$ (nM) | 47.4±2.6* | 3500±700* | 3700±200* |
| $K_d^{ens(EC+EO)}$ (nM) | 17.0±3.0# | 4300±900 | 4100±500* |
| $K_d^{ens(BC+EC)}$ (nM) | 7000±3400# | 7100±900 | 8500±1400 |
| $K_d^{EO}$ (nM) | 2.2±0.3# | 2.1±0.4 | 1.9±0.3* |
| $\Delta G^{BC}$ (kcal/mol) | −1.5±0.1 | — | — |
| $\Delta G^{EC}$ (kcal/mol) | −1.1±0.1 | −4.7±0.3 | −4.7±0.1 |
| $P^{BC}$ (%) | 64.3±6.6 | — | — |
| $P^{EC}$ (%) | 31.1±6.6 | 99.97±0.01 | 99.97±0.01 |
| $P^{EO}$ (%) | 4.6±0.7 | 0.03±0.01 | 0.03±0.01 |
| $\Delta G_{conf}^{E}$ (kcal/mol) | 0.4±0.2 | — | — |
| $\Delta G_{conf}^{O}$ (kcal/mol) | 1.1±0.1 | 4.7±0.3 | 4.7±0.1 |
| $\Delta G_{conf}^{E\ tot}$ (kcal/mol) | 0.3±0.2 | — | — |
| $\Delta G_{conf}^{Activation}$ (kcal/mol) | 1.7±0.1 | 4.7±0.3 | 4.7±0.1 |

[a]Affinity data from Fig 3A
[b]Affinity data from Panel B
[c]Affinity data from Fig 3B

**D** Clasped complex N-glycan $\alpha_5\beta_1$ ectodomain

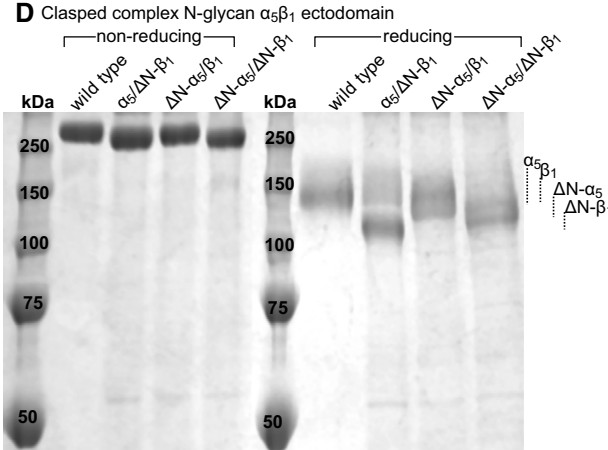

**E** Clasped complex N-glycan $\alpha_5\beta_1$ ectodomain[a]

| | Wild type | ΔN-$\alpha_5$/$\beta_1$ | $\alpha_5$/ΔN-$\beta_1$ | ΔN-$\alpha_5$/ΔN-$\beta_1$ |
|---|---|---|---|---|
| $K_d^{ens(Basal)}$ (nM) | 24.0±3.5* | 45.6±5.0 | 31.6±3.9 | 72.1±2.2* |
| $K_d^{ens(EC+EO)}$ (nM) | 3.6±0.4* | 3.2±0.2 | 5.7±0.6 | 5.4±0.4* |
| $K_d^{ens(BC+EC)}$ (nM) | 5700±3100 | ND | ND | ND |
| $K_d^{EO}$ (nM) | 2.6±0.7* | 1.8±0.2 | 2.0±0.3 | 2.3±0.7* |
| $\Delta G^{BC}$ (kcal/mol) | −1.2±0.2 | −1.8±0.1 | −1.5±0.1 | −2.0±0.2 |
| $\Delta G^{EC}$ (kcal/mol) | 0.6±0.6 | 0.1±0.2 | −0.4±0.2 | −0.2±0.3 |
| $P^{BC}$ (%) | 85.1±2.7 | 93.1±0.9 | 82.0±2.9 | 92.6±0.6 |
| $P^{EC}$ (%) | 4.2±3.4 | 3.0±0.7 | 11.7±2.6 | 4.3±1.1 |
| $P^{EO}$ (%) | 10.8±3.3 | 3.9±0.6 | 6.3±1.2 | 3.2±1.0 |
| $\Delta G_{conf}^{E}$ (kcal/mol) | 1.8±0.5 | 2.0±0.1 | 1.1±0.1 | 1.8±0.2 |
| $\Delta G_{conf}^{O}$ (kcal/mol) | −0.6±0.2 | −0.1±0.2 | 0.4±0.2 | 0.2±0.3 |
| $\Delta G_{conf}^{E\ tot}$ (kcal/mol) | 1.0±0.1 | 1.5±0.1 | 0.9±0.1 | 1.5±0.1 |
| $\Delta G_{conf}^{Activation}$ (kcal/mol) | 1.2±0.2 | 1.9±0.1 | 1.6±0.1 | 2.0±0.2 |

[a]Affinity data from Appendix Fig S4

**F** Clasped complex N-glycan $\alpha_5\beta_1$ ectodomain

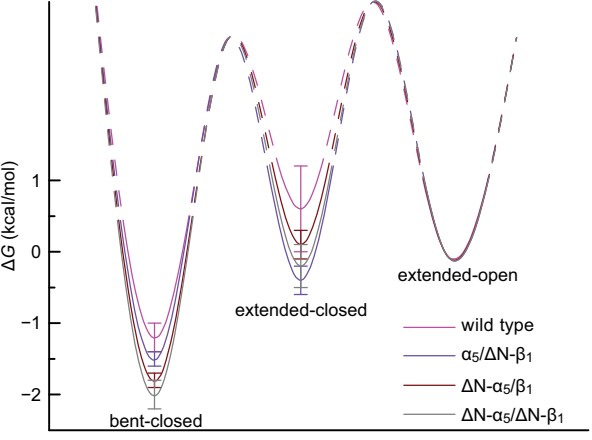

Figure 6.

## Discussion

### Methodologies for measuring intrinsic affinities and conformational equilibria of cell surface receptors

We have advanced here an approach to measuring conformational equilibria of multi-domain, multi-subunit receptors that should be extendible to many other cell surface receptors. Using antibodies to stabilize the same conformational states that proteins exhibit in the absence of antibodies is supported by the finding that antibodies usually stabilize proteins in conformations near energy minima (Kastritis et al, 2011). Indeed, one of the antibodies studied here, SG/19, binds to a closed headpiece conformation indistinguishable in atomic structures from that in the absence of Fab (Nagae et al, 2012; Xia & Springer, 2014). Thermodynamic studies using Fabs were validated here in a study without Fabs; complex N-glycans were shown with EM to stabilize the EO conformation. However, great care needs to be taken to validate the assumption that Fab-stabilized states resemble those in native proteins, and we have already found one exception, with TS2/16 Fab (Su et al, 2016). Here we have used between two to four independent antibodies or antibody combinations, often binding to distinct domains, to stabilize ensembles containing the EO, EC, EO+EC, and EC+BC states. In all cases, independent antibodies that stabilized the same conformational state(s) as determined by EM yielded similar ensemble $K_d$ values, validating stabilization of the same state(s), and supporting the assumption that these states resembled those in the absence of Fab. Quantitatively, the antibodies must be highly state-specific in order to give large shifts in affinities, to give consistent intrinsic affinities on constructs with large differences in basal affinities, and to give similar intrinsic affinities using Fabs to distinct epitopes. Exceptions to the rule that antibodies stabilize native states might occur when a non-native Fab-bound state can be reached that is lower in energy than the Fab-bound native state. This appeared to occur with the $\Delta$N-$\alpha_5$/$\Delta$N-$\beta_1$ mutant in the presence of SG/19 Fab and ligand. Increasing concentrations of SG/19 Fab lowered affinity to a plateau value, but not to the same value as for wild-type $\alpha_5\beta_1$ or $\Delta$N-$\alpha_5$/$\Delta$N-$\beta_1$ with mAb 13. Our interpretation is that removal of N-linked sites nearby the SG/19 binding site enabled a non-native state in which the $\beta$I domain was not fully in the closed conformation and could bind ligand. This emphasizes the importance of the use of independent Fabs to stabilize the same state. We also re-iterate the importance of using monovalent Fabs rather than bivalent IgGs and of

using Fabs at concentrations well above their $EC_{50}$ values to ensure saturation of the ensemble with the desired conformation(s).

In passing, we mention two common fallacies in the integrin field. On "resting" cells or in the case of purified integrins in the absence of activation, the low-affinity conformation is not uniformly adopted, as shown by our ensemble measurements. Furthermore, $Mn^{2+}$ does not "maximally" activate integrins, as shown by their increased affinity in the presence of Fabs that stabilize the EO state. In the absence of stabilizing Fabs, integrins are present in all three conformations in all conditions we examined. The ensemble may contain all three conformations in similar amounts, as for the unclasped ectodomain with complex glycosylation, or one conformation in great excess over the others, as for the BC conformation on cell surfaces.

### Intrinsic affinities of integrin states

A key question in the integrin field has been how adhesiveness and affinity are regulated. However, elucidation of integrin affinity regulation has been elusive (Schürpf & Springer, 2011; Zhu et al, 2013). Understanding how integrins bind ligand and connect to the cytoskeleton to mediate adhesion and cell migration requires not only the parts list and how the parts fit together, but also quantitative understanding of the binding affinities of those parts and the equilibria that regulate integrin conformation and affinity. Here, we have achieved such an understanding for an integrin of central importance in cell migration, signaling, adhesion, and assembly of fibronectin into the fibrils characteristic of the chordate extracellular matrix (Schwarzbauer & DeSimone, 2011). Our studies now make the biological processes in which integrin $\alpha_5\beta_1$ participates accessible to quantitative methods that seek to relate biological inputs to biological outputs in living cells.

Previous studies have correlated integrin adhesiveness and high affinity for ligand with the open conformation of the headpiece (Takagi et al, 2002, 2003; Xiao et al, 2004; Chen et al, 2010; Schürpf & Springer, 2011; Zhu et al, 2013). However, neither the affinities for ligand intrinsic to the closed and open headpiece conformations nor the magnitude of the affinity increase were previously measured. Here, we have measured intrinsic affinities of extended-open $\alpha_5\beta_1$ for RGD peptide, cRGD, and Fn3$_{9-10}$ as 71, 2.2, and 0.44 nM, respectively. We further determined the intrinsic affinity of closed $\alpha_5\beta_1$ for cRGD as 7,000–8,000 nM, and for Fn3$_{9-10}$ as 2,700–2,900 nM. The large magnitude of the affinity increase of

**Figure 7. Conformational equilibria and intrinsic affinity of intact $\alpha_5\beta_1$ on cell surfaces.**
Mean fluorescence intensity (MFI) after background subtraction is from quantitative fluorescence flow cytometry of K562 cells without washing.

A  Determination of $EC_{50}$ values for conformation-selective Fabs from enhancement of 10 nM Alexa488-Fn3$_{9-10}$ binding and fitting to Appendix Equation S2. Mean ± s.d. of least square fits to triplicates.
B  Affinity of $\alpha_5\beta_1$ for Alexa488-Fn3$_{9-10}$ in the presence of indicated Fabs.
C  Affinity of $\alpha_5\beta_1$ for Fn3$_{9-10}$ by enhancement of 0.4 nM Alexa488-12G10 Fab binding.
D  Affinities of $\alpha_5\beta_1$ for Alexa488-12G10 Fab in the presence of indicated Fabs.
E  Thermodynamics and intrinsic affinities of $\alpha_5\beta_1$ conformational states on K562 cells and Jurkat cells (Appendix Fig S8). Affinity for Fn3$_{9-10}$ of the closed conformations of $\alpha_5\beta_1$ on K562 cells was estimated from $K_d^{EO}$ using the same fold-difference as found with Fn3$_{9-10}$ for the $\alpha_5\beta_1$ ectodomain (Fig 4D). Since 12G10 stabilizes the open conformation only, thermodynamic calculations use $K_a = 0$ for the closed conformations.
F  Energy landscape plots, as described in the Fig 4E legend, comparing K562 $\alpha_5\beta_1$ to the clasped and unclasped $\alpha_5\beta_1$ ectodomain with complex glycosylation (Fig 5B).
G  Summary of the intrinsic affinities of $\alpha_5\beta_1$ conformational states on K562 cells and comparison of the conformational equilibria for $\alpha_5\beta_1$ on K562 cells and for the unclasped $\alpha_5\beta_1$ ectodomain with complex glycosylation.

Data information: (B–F) Error definitions are as described in the legend to Fig 4.

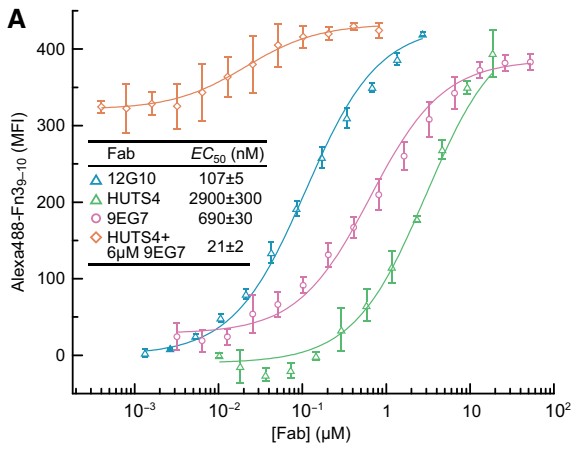

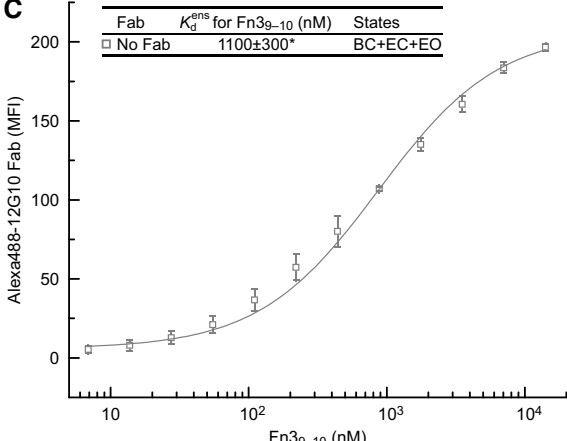

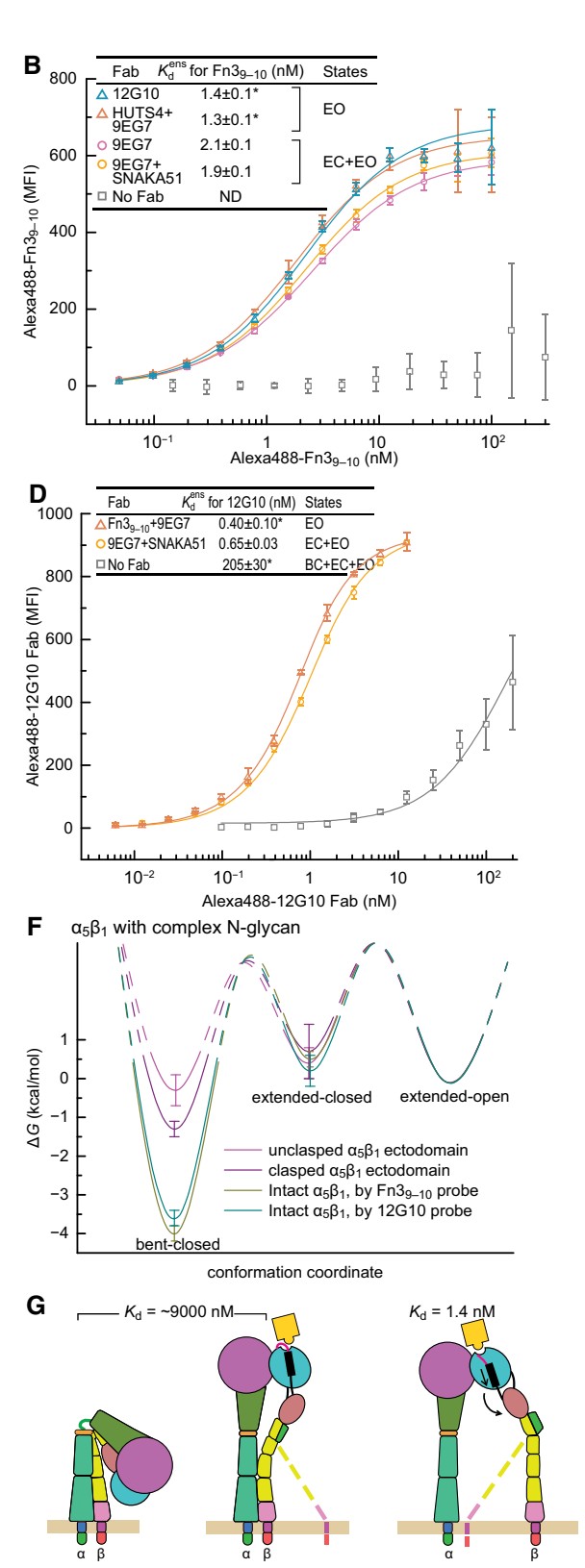

**E** Intact α5β1 on cell surface

| Cell type | K562 | | Jurkat |
|---|---|---|---|
| Ligand or Fab | Fn3 9–10 [a] | 12G10 [b] | Fn3 9–10 [c] |
| $K_d^{ens(Basal)}$ (nM) | 1100±300* | 205±30* | 750±60* |
| $K_d^{ens(EC+EO)}$ (nM) | 2.0±0.1# | 0.65±0.03 | 1.7±0.2 |
| $K_d^{ens(BC+EC)}$ (nM) | ~9000 | NA | ~9000 |
| $K_d^{EO}$ (nM) | 1.4±0.1# | 0.40±0.10* | 1.6±0.2 |
| $\Delta G^{BC}$ (kcal/mol) | −4.0±0.2 | −3.6±0.2 | −3.7±0.1 |
| $\Delta G^{EC}$ (kcal/mol) | 0.5±0.2 | 0.3±0.4 | 1.6±1.7 |
| $P^{BC}$ (%) | 99.84±0.05 | 99.68±0.05 | 99.82±0.03 |
| $P^{EC}$ (%) | 0.05±0.02 | 0.12±0.05 | 0.01±0.03 |
| $P^{EO}$ (%) | 0.11±0.04 | 0.20±0.06 | 0.17±0.03 |
| $\Delta G_{conf}^E$ (kcal/mol) | 4.4±0.2 | 3.9±0.3 | 5.3±1.6 |
| $\Delta G_{conf}^O$ (kcal/mol) | −0.5±0.2 | −0.3±0.4 | −1.6±1.7 |
| $\Delta G_{conf}^{E,tot}$ (kcal/mol) | 3.7±0.2 | 3.3±0.1 | 3.7±0.1 |
| $\Delta G_{conf}^{Activation}$ (kcal/mol) | 4.0±0.2 | 3.6±0.2 | 3.7±0.1 |

[a]Affinity data from Panel B&C
[b]Affinity data from Panel D
[c]Affinity data from Panel C&D in Appendix Fig S8

**Figure 7.**

4,000-fold to 6,000-fold should put to rest any lingering doubt in the integrin community about the relevance of changes in conformation and affinity to the regulation of integrin function and adhesiveness (Zhu *et al*, 2013).

## Conformational equilibria in integrin allostery

Our studies extend structural observations on the three overall integrin states in EM (Takagi *et al*, 2002, 2003; Zhu *et al*, 2008; Chen *et al*, 2010; Su *et al*, 2016) by showing that these correspond to three states with discrete free energies. The bent-closed conformation of integrins is compact, well defined, and amenable to crystallography as shown with integrins $\alpha_V\beta_3$, $\alpha_{IIb}\beta_3$, and $\alpha_X\beta_2$ (Xiong *et al*, 2001; Zhu *et al*, 2008; Xie *et al*, 2010). However, extended-closed and extended-open conformations show flexible lower legs and have thus far been amenable to EM and small-angle X-ray scattering but not crystallography; atomic understanding of the headpiece and $\beta$-legs outside of the bent conformation comes from crystallization of ectodomain fragments (Springer & Dustin, 2012). EM and crystal structures show that the $\beta$-leg is flexible, particularly at the knee between EGF1 in the upper leg and EGF2 in the lower leg (Shi *et al*, 2007). Flexion of the $\alpha$-subunit occurs at its knee between thigh and calf-1 and also at the $\beta$-propeller interface with thigh (Xie *et al*, 2010). Because of the considerable flexibility of the integrin legs when extended, the extended-closed and extended-open states should be considered not single but overall states, each with a large number of microstates that vary in leg domain orientation (Takagi *et al*, 2002).

With this heterogeneity among microstates in mind, it is gratifying that the studies here show that the bent-closed, extended-closed, and extended-open states defined structurally are also distinct in free energy. Thus, Fabs that induced extension by binding to the interface in the $\alpha$-subunit lower leg between the calf-1 and calf-2 domains, to the PSI domain in the upper $\beta$-leg, or to the I-EGF2 domain in the lower $\beta$-leg, induced extended states indistinguishable in free energy. The difference in energy between the BC and EC states may relate to breakage of the large interfaces in the BC conformation between the headpiece and lower legs and between the $\alpha$-subunit and $\beta$-subunit legs (Takagi *et al*, 2002), the replacement of these interfaces by solvent in the extended-closed conformation, and the large number of leg conformations accessible after extension. By contrast, differences between interfaces to which Fabs bind and among the conformational microstates accessible in the two extended overall states with different bound Fab must be minor in comparison.

Headpiece opening involves highly discrete $\beta$I domain $\alpha$7-helix pistoning and remodeling of a large $\beta$I/hybrid domain interface in the closed conformation to a smaller interface in the open conformation (Xiao *et al*, 2004). Again, binding of Fabs to distinct epitopes in the $\beta$I and hybrid domains suggested an extended-open state with discrete $\Delta G$ despite evidence for leg flexibility in this state. Furthermore, the combined use of two Fabs to induce extension and opening compared to the use of one open-stabilizing Fab resulted in no significant affinity differences compared to the scale of the affinity differences between states.

Measurements of $\alpha_5\beta_1$ conformational equilibria on the K562 erythroleukemia cell line quantify the energy requirements in the pathway of integrin activation from the bent-closed to the extended-open conformation. Surprisingly, energy is only required for extension;

once the integrin extends, the open headpiece is slightly energetically favored over the closed headpiece. Furthermore, in the absence of ligand the basal integrin ensemble on K562 cells is 99.76 ± 0.08% bent-closed, 0.09 ± 0.04% extended-closed, and 0.15 ± 0.03% extended-open. In K562 cells, the energy input required to stabilize integrin extension and headpiece opening of ~4 kcal/mol may come from the ATP hydrolysis required to drive actin polymerization into filaments and myosin-dependent actin filament contraction, and be coupled to integrin conformational change through the cytoskeletal force model of integrin activation (Zhu *et al*, 2008).

## $\alpha_5\beta_1$ energy landscape on the cell surface

Comparisons among intact $\alpha_5\beta_1$ on K562 cells and soluble fragments provide insights into the molecular components that contribute to the energy landscape. On K562 cells, extension requires a large energy input; in contrast, headpiece opening spontaneously follows extension because the EO conformation is lower in energy than the EC conformation. To estimate the contribution to energetics of the TM and cytoplasmic domains, as a first approximation the energy required for extension of the ectodomain portion of $\alpha_5\beta_1$ on the cell surface and the unclasped $\alpha_5\beta_1$ ectodomain with complex N-glycans in solution may be compared. This comparison suggests that a substantial amount of energy is required for alterations in the TM and cytoplasmic domains that are associated with extension. Thus, inside-out signaling must input, through alterations in TM/cytoplasmic domains, about 3.5 kcal/mol ($\Delta G_{conf}^{E\,tot}$) to convert the EC+EO conformations from a population of 0.2% in the basal ensemble to a population of 50% in an active ensemble. Considerable evidence shows that both the $\alpha$- and $\beta$-subunit cytoplasmic domains and transmembrane domains separate upon integrin activation (Kim *et al*, 2003; Luo *et al*, 2004a, 2005; Zhu *et al*, 2009). Structures determined for the TM domains in intact integrins on cell surfaces and for TM domain peptides in bicelles show an interface between the TM domains that extends through the bilayer and is enhanced by a reverse-turn at a GFFKR motif at the cytoplasmic face (Lau *et al*, 2009; Zhu *et al*, 2009). The current study has not probed the energetics of integrin TM and cytoplasmic domain association; however, the large contribution to the energetics of extension ($\Delta G_{conf}^{E}$) by the TM and cytoplasmic domains suggests that on cell surfaces, a higher proportion of EC state than BC state integrins may have their $\alpha$- and $\beta$-subunit TM and cytoplasmic domains dissociated from one another. On cells, integrins associate with cytoplasmic proteins that stabilize activation or inactivation (Bouvard *et al*, 2013; Calderwood *et al*, 2013), and these interactions will also contribute to the free energies we measure at 20–25°C where the membrane bilayer is fluid. We found no difference in conformational state free energy between $\alpha_5\beta_1$ on two different cell types; however, such differences may be found for other cell types or other integrins.

Relief of crowding or repulsive interactions between the lower $\alpha$- and $\beta$-subunit legs is an important driver of headpiece opening and may also drive TM domain separation. Compared to the ectodomain construct, much more energy is required for headpiece opening in protein constructs lacking one or both lower legs. In the bent-closed conformation on cell surfaces, the lower legs meet the upper legs at a point where the integrin $\alpha$ and $\beta$ knees are close together. The lower legs also meet, through short linkers of ~8 residues in $\alpha_5$ and

~4 residues in $\beta_1$, at the $\alpha$ and $\beta$ TM domains where they are close together (Zhu *et al*, 2009). Therefore, crowding interactions between the lower $\alpha$ and $\beta$ legs may drive TM/cytoplasmic domain separation similarly to headpiece opening.

## Measurements of equilibria provide insights into integrin structure and function orthogonal but complementary to insights from structural biology

Our findings that the extended-open conformation of the ectodomain is stabilized by (i) the presence of the lower legs and (ii) large N-glycans suggest the hypothesis that in multi-domain receptors, non-ligand-binding (leg) domains and N-glycans may have previously unappreciated roles in regulating affinity for ligands and conformational equilibria. In the case of integrins, the $\alpha$- and $\beta$-subunit knees are 10 Å apart in the bent-closed conformation and 100 Å apart in the extended-open conformation (Springer *et al*, 2008; Zhu *et al*, 2008). Thus, in the extended-closed conformation, the lower integrin $\alpha$- and $\beta$-legs are very close at the knees and will crowd or repel one another. This hypothesis predicts that both integrin lower legs must be present to obtain crowding or repulsion and relief by opening, as we experimentally verified.

Conformational equilibria and allosteric regulation were elegantly described decades ago for membrane channels using measurements of their opening and closing, and more recently measured for G protein-coupled receptors (Ruiz & Karpen, 1997; Horrigan *et al*, 1999; Lape *et al*, 2008; Park *et al*, 2008; Cecchini & Changeux, 2015; Manglik *et al*, 2015). All of these receptors have a large domain composed of multiple TM domains embedded in the membrane and have N-glycosylated extracellular loops or domains. Another large class of glycosylated receptors to which integrins belong have extracellular domains in tandem with single-span TM domains. Many of these receptors bind ligands in membrane distal domains and have a larger number of domains, often cysteine-rich, immunoglobulin-like or fibronectin type 3-like, that link the ligand-binding domain(s) to single TM domains (Fig 1D). We propose that the glycosylated tandem domains that link ligand-binding and TM domains in such receptors are analogous to the leg domains of integrins. The active states of integrins and these receptors have an inverse relationship, such that the active, open, TM domain-apart integrin state corresponds to the monomeric, inactive state of cytokine and growth factor receptors, whereas the inactive, bent-closed, TM domain-together state of integrins corresponds to the active, multimeric, ligand-bound, TM domain-together state of such receptors (Fig 1D). Such receptors do not signal in the absence of cross-linking by ligand, and it will be interesting to investigate whether inter-monomer crowding or repulsive interactions between leg domains are also present in cytokine and growth factor receptors and function to prevent multimerization and signaling in the absence of ligand. We propose that glycans and leg domains may regulate monomer-dimer equilibria in such receptors similarly to their regulation of $K_{conf}^{Activation}$ in integrins (Fig 1).

The large amount of space that N-glycans occupy on the surface of receptors (Fig 1B and D) is often not recognized. N-glycans are often removed prior to crystallization. Moreover, glycosidic bonds are typically free to rotate, both at linkages to Asn and monosaccharide-monosaccharide linkages, and thus, glycan residues are usually difficult to visualize in crystal structures even when

present. This flexibility enables N-glycans on one domain to sweep out a large hydrodynamic radius and to crowd or repel with their sialic acid residues other protein domains or N-glycans linked to other sites. The composition and chemical structures of the N-glycans in native integrin $\alpha_5\beta_1$ are well defined (Sieber *et al*, 2007), enabling us to create a scale model comparing the size of the protein and carbohydrate components of $\alpha_5\beta_1$ (Fig 1B). Building a natively glycosylated model of $\alpha_5\beta_1$ BC was challenged by the difficulty of obtaining orientations of glycosidic linkages that prevented N-glycans from clashing with protein or nearby N-glycans. Much of the protein surface in integrins (Fig 1A, lower) is obscured by N-glycans (Fig 1B). Once an integrin extends, the size, flexibility, and hydration of N-glycans creates potential overlap or repulsion between N-glycans attached to different domains, and the protein domains in the lower legs have the potential themselves to also sterically overlap. Such interactions might explain the contribution of the N-glycans and lower leg domains to favoring the less compact EO state over the more compact BC and EC states in integrins. We believe that the effects of glycan shortening and N-linked site removal are general rather than associated with specific N-linked sites. We have used no structural information to guide removal of N-linked sites in $\alpha_5$ and $\beta_1$, and the sites removed (white glycans, Fig 1B) have a similar distribution to sites that remained (blue glycans, Fig 1B).

We were surprised by the large effect of N-glycans and leg domains on conformational equilibria. Compared to shaved glycans, complex glycans raise affinity for ligand of the basal $\alpha_5\beta_1$ ectodomain ensemble by 8- to 10-fold, much more than clasp removal. Furthermore, the presence of both lower legs raises ensemble affinity of the ectodomain by 80-fold compared to the headpiece.

Thus, N-glycans and the lower integrin legs each have major roles in regulating integrin allostery. Although the $\beta$I/hybrid and I-EGF1/I-EGF2 interfaces also may regulate integrin allostery (Xiao *et al*, 2004; Smagghe *et al*, 2010), domains in proteins and their interfaces have many evolutionary constraints. The identity and number of upper and lower leg domains in integrin $\alpha$- and $\beta$-subunits are invariant in multi-cellular animals from sponges to chordates and thus have been fixed in evolutionary history for ~600 million years. This invariance likely reflects how closely the legs pack in the bent conformation and the precise requirements of integrins as molecular machines (Takagi *et al*, 2002). While integrin legs have previously been viewed largely as passive conduits for transmission of signals between the ligand-binding head and the plasma membrane, our study demonstrates that the lower legs have a key role in regulating integrin allostery, and thus activation.

Our results establish the principle that variation in N-glycosylation site number can regulate conformational equilibria of multi-domain receptors. N-glycosylation is not only a previously unrecognized mechanism for regulating conformational equilibria but is also evolutionarily facile. We have shown that a decrease in the number of N-glycosylation sites on integrin $\alpha_5\beta_1$ stabilizes its bent-closed and extended-closed conformations and lowers the ligand-binding affinity of its basal ensemble; decreases in the number and complexity of carbohydrate residues at each N-linked site had a similar effect. Among integrins, $\beta_1$ integrins are widely expressed in extravascular environments and are considered to be basally active, whereas $\beta_2$ and $\beta_3$ integrins are expressed in vascular environments and are considered basally inactive. Our study opens up

testing the hypothesis that integrins differ in their activation set points, and provides benchmark measurements on $\alpha_5\beta_1$. It is interesting that among integrin $\beta$-subunits, $\beta_1$ has the most N-glycosylation sequons (12), $\beta_4$ has the least (5), and $\beta_2$ and $\beta_3$ are tied for next least (6). Variation in the number of N-glycans is also great among integrin $\alpha$-subunits and ranges from 26 on $\alpha_1$ to 5 on $\alpha_7$ and $\alpha_{IIb}$. Further study is required to determine the molecular basis for apparently markedly different set points among integrins in their basal activity on cells (Bazzoni *et al*, 1998). However, our studies raise the possibility that N-glycosylation site number may be among the mechanisms that regulate integrin basal activity and that N-glycosylation may play a wide role in regulating conformational equilibria of extracellular and membrane proteins.

# Materials and Methods

### Fabs

Briefly, sources of hybridomas for 12G10, HUTS4, 8E3, 9EG7 and SNAKA51 (Askari *et al*, 2010), and SG/19 and TS2/16 (Luo *et al*, 2004b) were as described in the citations. N29 (Ni *et al*, 1998) and mAb13 (Akiyama *et al*, 1989) hybridomas were kind gifts of J. Wilkins (U. Manitoba, Canada) and K. Yamada (NIH, USA), respectively. Anti-$\alpha_5\beta_1$ hybridoma sources and details of purification of IgG by protein G affinity and preparation of Fab fragments with papain digestion and purification of Fabs by Hi-Trap Q chromatography were as described in detail (Su *et al*, 2016).

### Integrin $\alpha_5\beta_1$ constructs

DNA constructs, stable HEK 293S GnTI$^{-/-}$ cell lines (*N*-acetylglucosaminyltransferase I deficient), and high-mannose $\alpha_5\beta_1$ ectodomain ($\alpha_5$ F1 to Y954 and $\beta_1$ Q1 to D708), headpiece ($\alpha_5$ F1 to L609 and $\beta_1$ Q1 to E481), and semi-truncated ($\alpha_5$ F1 to L609 and $\beta_1$ Q1 to D708) fragments were prepared as described (Takagi *et al*, 2001; Xia & Springer, 2014). High-mannose glycoforms were shaved with endoglycosidase H in 50 mM MES buffer (pH 5.6), 100 mM NaCl, 1 mM CaCl$_2$, and 1 mM MgCl$_2$, at an equal protein and enzyme mass ratio, for 12 h at room temperature, followed by gel filtration purification.

The complex glycoform of the $\alpha_5\beta_1$ ectodomain and its N-linked glycosylation site mutants with mature Asn residues at 256, 266, 483, 489, 568, and 634 in $\alpha_5$ ($\Delta$N-$\alpha_5$), or 74, 77, 343, 386, and 500 in $\beta_1$ ($\Delta$N-$\beta_1$) mutated to Arg were produced by co-transfecting HEK 293 cells with codon-optimized $\alpha_5$ and $\beta_1$ cDNAs with secretion peptide, purification tags, and C-terminal clasp (Takagi *et al*, 2001; Xia & Springer, 2014) in pcDNA3.1/Hygro(−) and pIRES vectors. Stable transfectants were selected with hygromycin (100 µg/ml) and G418 (1 mg/ml), and $\alpha_5\beta_1$ glycoproteins were purified as for the high-mannose glycoform.

### Peptide ligands and Fn3$_{9-10}$

Cyclic RGD peptide (ACRGDGWCG) and RGD peptide (GRGDSPK) (> 95% pure) were synthesized and labeled with FITC at the N-terminus via a 6-aminohexanoic acid spacer by GenScript (Piscataway, NJ). Human Fn3$_{9-10}$ (mature residues G1326 to T1509) and its S1417C mutant were expressed in *Escherichia coli* and purified as described (Takagi *et al*, 2001). The mutant was fluorescently labeled with Alexa Fluor 488 C5 maleimide (ThermoFisher Scientific) in PBS.

### Fluorescence polarization

Each sample (10 µl) contained 150 mM NaCl, 1 mM CaCl$_2$, 1 mM MgCl$_2$, 5 nM FITC-cRGD or FITC-RGD, $\alpha_5\beta_1$, and indicated Fabs and/or Fn3$_{9-10}$ in 20 mM Tris buffer (pH 7.4). For competitive binding, $\alpha_5\beta_1$ was pre-equilibrated with Fn3$_{9-10}$ for 2 h before mixing with FITC-cRGD. The mixture was allowed to equilibrate for 2 h (24 h with 12G10 Fab to reach equilibrium) before recording FP on a Synergy NEO HTS multi-mode microplate reader (Biotek).

### Quantitative fluorescent flow cytometry

K562 and Jurkat cells ($10^6$ cells/ml in RPMI-1640 medium, 10% FBS) were washed twice with assay medium (Leibovitz's L-15 medium, 10 mg/ml BSA) containing 5 mM EDTA, twice with assay medium, and resuspended in assay medium. Each sample (50 µl) contained cells ($2 \times 10^6$ cells/ml), Alexa488-Fn3$_{9-10}$ or Alexa488-12G10, and indicated Fabs or Fn3$_{9-10}$ in assay medium. The mixture was allowed to equilibrate for 1.5 h before flow cytometry (BD FACSCanto II) without washing (Chigaev *et al*, 2001; Dong *et al*, 2014). Alexa488-Fn3$_{9-10}$ or Alexa488-12G10 binding was measured as mean fluorescence intensity (MFI); background MFI was subtracted and was measured by the same method in the presence of 10 mM EDTA (K562) or 100-fold excess of $\alpha_5$-specific, ligand-blocking antibody mAb16 (Burrows *et al*, 1999) over Fn3$_{9-10}$ (Jurkat).

### Thermodynamic properties

Ligand-binding affinities of $\alpha_5\beta_1$ ensembles in the absence or presence of Fabs were measured by saturation binding or competitive binding using FP or flow cytometry. Probabilities of each $\alpha_5\beta_1$ conformation in the basal ensemble and their free energies relative to that of the extended-open conformation ($\Delta G^{BC}$ and $\Delta G^{EC}$; $\Delta G^{EO} = 0$) were calculated from intrinsic and ensemble affinities (Fig 1C). Appendix Equations S1–S94 used to obtain reported values are described in Appendix Supplementary Materials and Methods. Briefly, $K_d^{app}$ values were obtained from fits to Appendix Equations S16, S17, S26, and S28; $K_d^{ens}$ values in the presence of closure-stabilizing Fabs were calculated using Appendix Equation S64; in all other conditions, $K_d^{ens} = K_d^{app}$ (Appendix Fig S2); $P$ and $\Delta G$ values are calculated from $K_d^{ens}$ values with Appendix Equations S73–S77 and S82–S84, respectively, and their errors are calculated by propagation from $K_d^{ens}$ errors.

**Expanded View** for this article is available online.

### Acknowledgements

We thank Thomas Holder of Schrodinger for Pymol/Python ellipsoid and torus scripts. This work was supported by NIH grants P01-HL-103526 and R01-HL-131729, and by a Susan G. Komen Basic/Translational postdoctoral fellowship.[#]

[#]Correction added on 8 February 2017, after first online publication: the Acknowledgements section has been updated.

## Author contributions

JL, CL, and TAS designed research; JL, YS, WX, and YQ performed research; JL, YS, and TAS analyzed data and wrote the manuscript; MJH, DV, and CC provided reagents and aided in interpretation of results.

## Conflict of interest

The authors declare that they have no conflict of interest.

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
