## [Review Process File · The EMBO Journal]

Manuscript EMBO-2016-95803

Conformational Equilibria and Intrinsic Affinities Define Integrin Activation

Jing Li, Yang Su, Wei Xia, Yan Qin, Martin J. Humphries, Dietmar Vestweber, Carlos Cabañas, Chafen Lu and Timothy A. Springer

Corresponding author: Timothy Springer, Boston Children's Hospital/Harvard Medical School

Review timeline:

Submission date:	27 September 2016
Editorial Decision:	27 September 2016
Revision received:	15 November 2016
Editorial Decision:	07 December 2016
Revision received:	12 December 2016
Accepted:	15 December 2016

Editor: Ieva Gailite

Transaction Report:

1st Editorial Decision

27 September 2016

Thank you for submitting your manuscript for consideration by the EMBO Journal. It has now been seen by three referees whose comments are shown below.

As you can see from the comments, all three referees express interest in the presented analysis of integrin conformational state changes. However, they also raise significant concerns with the analysis that would have to be addressed in order to consider publication here. I would like to invite you to submit your revised manuscript while addressing the comments of all three referees, and focusing in particular on the following points:

1. All referees point out that the manuscript has to be rewritten to make it more accessible to general audience.
2. Validation of the specificity of the used antibodies to specific conformational states.
3. Validation of the quantification of integrin conformational state distribution in an alternative cell line.
4. Address the concerns regarding the interpretation of intramolecular crowding experiments.

I should add that it is EMBO Journal policy to allow only a single major round of revision, and acceptance of your manuscript will therefore depend on the completeness of your responses in this revised version.

When preparing your letter of response to the referees' comments, please bear in mind that this will form part of the Review Process File, and will therefore be available online to the community. For

more details on our Transparent Editorial Process, please visit our website:
http://emboj.embopress.org/about#Transparent_Process

Please feel free to contact me if have any further questions regarding the revision. Thank you for the opportunity to consider your work for publication. I look forward to your revision.

REFEREE REPORTS

Referee #1:

This manuscript describes a series of affinity measurements of ligand/integrin interactions using conformation-specific antibodies. A lot of work was done in this study. The affinities at different strengths correlate with the known conformational states of integrin, which have been extensively studied over the past 16 years. While the study does provide valuable information in understanding the conformation-dependent ligand-integrin interaction, the results were not written in a way that provide highly novel conclusion given the already widely appreciated conformation-specific functional states of integrin. A significant concern is how accurate these antibodies can detect specific conformational states. How can one know that a specific antibody only recognizes one state but not cross-react with another conformation in certain degree?

Other minor concerns:

1. The authors do not appear to consider the emerging negative regulators of integrin activation at all (Bouvard et al., 2013, Nat. Rev. Mol Cell Biol). In other words, when the authors measure the ligand affinity in cells such as K562 cells, how could they exclude the possibility that certain conformation of integrin in inactive (clapsed) state is not bound to an integrin inactivator such as filamin? In fact, it has been shown recently that filamin may bind and stabilize the low affinity integrin state (clapsed form) (Liu et al., Nat Struct Mol Biol., 2015). This should be at least discussed.
2. The manuscript is written densely and should be revised in more concise way to emphasize the major findings.

Referee #2:

The authors report on a very detailed study of binding affinities of $\alpha 5 \beta 1$ integrin to RGD peptides. Among the 24 human integrins, $\alpha 5 \beta 1$ integrin sticks out as the fibronectin receptor and its physiological relevance therefore is very high. Moreover, as the authors discuss extensively, it is not only a model for the integrin family, but also for cytokine and growth factor receptors. The authors use two different techniques to measure the affinities between receptors and ligands. For the most part of the manuscript (figures 2-6), they use fluorescence polarization (FP) with a commercial instrument on the soluble ectodomain of the integrin. To make contact with the rate equation approach, it is then assumed that the increase in FP is directly proportional to the increase in bound ligand. In the last part of the manuscript (figure 7), they use flow cytometry to measure affinity on the surface of a cancer cell line. Integrins are known to exist in three different conformations (extended open, EO; extended closed, EC; bent closed, BC) and the authors use the antibodies that have been raised earlier to select for these conformations to measure how affinities change as the conformational ensemble is shifted. Using the rate equation approach, affinities are converted into free energy differences. The main result is that the adhesive conformation, EO, is actually a high energy conformation, thus cellular energy (most likely from the cytoskeleton) is needed to populate this state, consistent with the idea of inside-out signaling to the integrins. In the solution studies with

the ectodomain, free energy decreases through EC to BC. In the cell studies, EC has a slightly higher energy than BC, reflecting the importance of the transmembrane and cytoplasmic domains, which provide additional binding interfaces and entropic contributions. In regard to numbers, the authors get very dramatic results, with a 5.000 fold higher affinity of EO in the solution experiment, but with an extremely low probability of occupation (0.15 percent) in the cell experiment. Another important aspect of this study is the role of N-glycans. By comparing wildtype, shaved and high mannose conditions, the authors demonstrate a very strong regulatory role of glycosylation.

In general, this study is very solid, detailed and interesting. It contains a wealth of novel information and is highly relevant for the integrin field and also for the surface receptor signaling field in general. However, the manuscript in its current state is written in a very compact form and not easy to digest. I suggest to better separate the results of this study from the interpretation, because this tends to be intermingled throughout the manuscript. The introduction should be rewritten and not jump directly to the affinity issue (Fig. 1C), but first give the general background and explain the different conformations (Fig. 1A). Although I like the glycan part, it is somehow an add-on to the main story and makes the manuscript even more complex; the authors should make a stronger effort to explain from the start what has been measured and why. The discussion is very much centered on open issues in the integrin field and could discuss in more details the results and limitations of this specific study.

Here is a list of limitations that I feel should be discussed in more detail.

1. 200.000 expressed $\alpha 5\beta 1$ integrins on the cell surface seems to be a very high number, an order of magnitude higher than I expected. What are typical number of non-transformed cells and how telling is this choice of cell type ? Here results for another cell type would be very helpful, as this part is relatively short anyway, but essential for the conclusion drawn.
2. Only 0.15 percent EO is also an extreme result and might be a consequence of the rate equation approach (see below). Is there an independent way to confirm this prediction ? What are the error bars on this statement ?
3. The rate equation approach looks reasonable to me, but one has to clearly state its limitations. It uses the equations for a non-interacting system (ideal gas) and a Boltzmann distribution for three distinct conformations. I do not agree that this study shows that these three conformations are thermodynamically separated, this is assumed from the very beginning and validated by consistency, but not be an actual experiment. Thus I suggest to remove this statement from the abstract and to give a more detailed discussion in the main text.
4. The crowding concept is very interesting but should also be explained in the introduction because it is mentioned in the abstract. Again this part is somehow speculative, because crowding usually implies some strong entropic component, which cannot be decided on the level of affinity experiments as done here. How can the authors distinguish between electrostatic repulsion and steric interactions between the legs ? This is a very interesting question that should be addressed in future studies, for example by mutation studies. Here the subject should be discussed with more care.

Referee #3:

This is an extremely interesting study on conformation-specific ligand binding affinities. The idea of using antibodies to trap receptors in specific conformations is powerful and should have wide applicability. I have a few minor comments that should help improve the already high quality of this paper.

1. Given the potential of conformation trapping by antibodies, the authors may want to comment on how their approach can be adapted for other receptors.
2. I'm not convinced of the "intramolecular" crowding hypothesis. If there are multiple copies of the integrin molecules in the membrane, then it's inevitable that these molecules will interact with each other, providing ample chance for "intermolecular" crowding. The physical picture of the latter scenario is simple and clear [see, e.g., Zhou, J. Phys. Chem. B 113, 7995-8005 (2009)], whereas the

"intramoelcular" scenario seems hard to imagine. In any event, if the authors insist they are correct, they should add a plausible physical picture for how it works.

3. p. 5, line 3, "affinity" -- I assume the authors mean affinity for the RGD ligand. If so, this should be explicitly stated. On first reading, I thought the authors were referring to affinities for the antibodies.

1st Revision - authors' response

15 November 2016

Thank you for the high quality reviews. We have revised the MS to address all reviewer concerns. It has been rewritten to provide more smooth transitions between sections and rationale. The crowding section has been rewritten and toned down. We now indicate it is either crowding or repulsion. We have moved discussion out of results. We have added a new section at the beginning of Discussion reprising our results and describing our validation methods and their limitations. We have also added measurements on a second type of cell. We note comments below in blue to all points of the review.

As you can see from the comments, all three referees express interest in the presented analysis of integrin conformational state changes. However, they also raise significant concerns with the analysis that would have to be addressed in order to consider publication here. I would like to invite you to submit your revised manuscript while addressing the comments of all three referees, and focusing in particular on the following points:

1. All referees point out that the manuscript has to be rewritten to make it more accessible to general audience.
2. Validation of the specificity of the used antibodies to specific conformational states.
3. Validation of the quantification of integrin conformational state distribution in an alternative cell line.
4. Address the concerns regarding the interpretation of intramolecular crowding experiments.

Referee #1:

This manuscript describes a series of affinity measurements of ligand/integrin interactions using conformation-specific antibodies. A lot of work was done in this study. The affinities at different strengths correlate with the known conformational states of integrin, which have been extensively studied over the past 16 years. While the study does provide valuable information in understanding the conformation-dependent ligand-integrin interaction, the results were not written in a way that provide highly novel conclusion given the already widely appreciated conformation-specific functional states of integrin. A significant concern is how accurate these antibodies can detect specific conformational states. How can one know that a specific antibody only recognizes one state but not cross-react with another conformation in certain degree?

We have now discussed cross-reactivity in the text. In first paragraph of Discussion we state "Quantitatively, the antibodies must be highly state-specific in order to give large shifts in affinities, to give consistent intrinsic affinities on constructs with large differences in basal affinities, and to give similar intrinsic affinities using Fabs to distinct epitopes."

Other minor concerns:

1. The authors do not appear to consider the emerging negative regulators of integrin activation at all (Bouvard et al., 2013, Nat. Rev. Mol Cell Biol). In other words, when the authors measure the ligand affinity in cells such as K562 cells, how could they exclude the possibility that certain conformation of integrin in inactive (clasped) state is not bound to an integrin inactivator such as filamin? In fact, it has been shown recently that filamin may bind and stabilize the low affinity integrin state (clasped form) (Liu et al., Nat Struct Mol Biol., 2015). This should be at least discussed.

We agree. We now cite the review by Bouvard D, Pouwels J, De Franceschi N, Ivaska J (2013). In Discussion we say "On cells, integrins associate with cytoplasmic proteins that stabilize activation or inactivation (ref), and these interactions will also contribute to the free energies we measure at 20-25° C where the membrane bilayer is fluid. We found no difference in free energy between $\alpha_5\beta_1$

on two different cell types; however, such differences may be found for other cell types or other integrins.”

2. The manuscript is written densely and should be revised in more concise way to emphasize the major findings.

We agree and have done so as described above.

Referee #2:

The authors report on a very detailed study of binding affinities of $\alpha 5 \beta 1$ integrin to RGD peptides. Among the 24 human integrins, $\alpha 5 \beta 1$ integrin sticks out as the fibronectin receptor and its physiological relevance therefore is very high. Moreover, as the authors discuss extensively, it is not only a model for the integrin family, but also for cytokine and growth factor receptors. The authors use two different techniques to measure the affinities between receptors and ligands. For the most part of the manuscript (figures 2-6), they use fluorescence polarization (FP) with a commercial instrument on the soluble ectodomain of the integrin. To make contact with the rate equation approach, it is then assumed that the increase in FP is directly proportional to the increase in bound ligand. In the last part of the manuscript (figure 7), they use flow cytometry to measure affinity on the surface of a cancer cell line. Integrins are known to exist in three different conformations (extended open, EO; extended closed, EC; bent closed, BC) and the authors use the antibodies that have been raised earlier to select for these conformations to measure how affinities change as the conformational ensemble is shifted. Using the rate equation approach, affinities are converted into free energy differences. The main result is that the adhesive conformation, EO, is actually a high energy conformation, thus cellular energy (most likely from the cytoskeleton) is needed to populate this state, consistent with the idea of inside-out signaling to the integrins. In the solution studies with the ectodomain, free energy decreases through EC to BC. In the cell studies, EC has a slightly higher energy than BC, reflecting the importance of the transmembrane and cytoplasmic domains, which provide additional binding interfaces and entropic contributions. In regard to numbers, the authors get very dramatic results, with a 5.000 fold higher affinity of EO in the solution experiment, but with an extremely low probability of occupation (0.15 percent) in the cell experiment. Another important aspect of this study is the role of N-glycans. By comparing wildtype, shaved and high mannose conditions, the authors demonstrate a very strong regulatory role of glycosylation.

In general, this study is very solid, detailed and interesting. It contains a wealth of novel information and is highly relevant for the integrin field and also for the surface receptor signaling field in general. However, the manuscript in its current state is written in a very compact form and not easy to digest. I suggest to better separate the results of this study from the interpretation, because this tends to be intermingled throughout the manuscript. The introduction should be rewritten and not jump directly to the affinity issue (Fig. 1C), but first give the general background and explain the different conformations (Fig. 1A). Although I like the glycan part, it is somehow an add-on to the main story and makes the manuscript even more complex; the authors should make a stronger effort to explain from the start what has been measured and why. The discussion is very much centered on open issues in the integrin field and could discuss in more details the results and limitations of this specific study.

We have done so as described above.

Here is a list of limitations that I feel should be discussed in more detail.

1. 200.000 expressed $\alpha 5 \beta 1$ integrins on the cell surface seems to be a very high number, an order of magnitude higher than I expected. What are typical number of non-transformed cells and how telling is this choice of cell type? Here results for another cell type would be very helpful, as this part is relatively short anyway, but essential for the conclusion drawn.

That number is well determined in the literature and we have moved the citation to clarify.

Transformed cells are larger than most counterparts. Platelets have 40,000 molecules per cell of integrin $\alpha_{IIb}\beta_3$, and they are far smaller than K562 cells. We had data on other cell types on hand and have now added a dataset on Jurkat T lymphoblastoid cells which has lower expression of integrin $\alpha_5\beta_1$. The results are almost identical and appear in Fig. 7E and Supplemental Fig. S8.

2. Only 0.15 percent EO is also an extreme result and might be a consequence of the rate equation approach (see below). Is there an independent way to confirm this prediction? What are the error bars on this statement?

This result is quite accurate and comes directly from the affinity of the basal ensemble on the cell surface compared to the intrinsic affinity of EO. Its easily calculated as $[EO]/([BC] + [EC] + [EO]) = \text{ensemble affinity}/EO \text{ affinity}$ with a correction for a small contribution of BC and EC to basal affinity. The error bars are in Fig. 7E for each of two independent determinations. We have now

added the errors for those two independent determinations to the Discussion. “Furthermore, in absence of ligand the basal integrin ensemble on K562 cells is 99.76±0.08% bent-closed, 0.09±0.04% extended-closed, and 0.15±0.03% extended-open.”

3. The rate equation approach looks reasonable to me, but one has to clearly state its limitations. It uses the equations for a non-interacting system (ideal gas) and a Boltzmann distribution for three distinct conformations. I do not agree that this study shows that these three conformations are thermodynamically separated, this is assumed from the very beginning and validated by consistency, but not be an actual experiment. Thus I suggest to remove this statement from the abstract and to give a more detailed discussion in the main text.

The Boltzmann distribution is not only applicable to an ideal gas system. In the statistical mechanics or thermodynamics view of protein conformational ensembles (which concentrate on the average or most probable behavior of the molecules), the frequency of individual states at a given temperature also follows this empirical distribution. Even though this relationship is not perfect, it is very common and has been used to study protein folding, protein allostery, and energy landscapes of protein conformational ensembles. John Kuriyan’s textbook “The Molecules of Life” is now used to teach Physical Chemistry to undergraduates using proteins and nucleic acids instead of the gas law. We believe that our states are thermodynamically separate in the sense normally used, as represented by landscape diagrams with valleys separated by peaks. In solution we have every right to expect that the integrins are non-interacting, and they behave similarly on cell surfaces.

However, our phrasing about thermodynamics was broad and could be interpreted differently from what we meant. We certainly don't have enthalpy or entropy measurements. We have toned down thermodynamic arguments by omitting the term thermodynamics, and changing our phrasing to be operational for what we observe, which is “the three conformational states of integrin $\alpha_5\beta_1$ have discrete free energies.” As requested, we have changed the statement in the Abstract and covered this in the Discussion.

4. The crowding concept is very interesting but should also be explained in the introduction because it is mentioned in the abstract. Again this part is somehow speculative, because crowding usually implies some strong entropic component, which cannot be decided on the level of affinity experiments as done here. How can the authors distinguish between electrostatic repulsion and steric interactions between the legs? This is a very interesting question that should be addressed in future studies, for example by mutation studies. Here the subject should be discussed with more care. As requested, we added to Introduction a paragraph that begins “By measuring conformational equilibria here, we have also discovered that previously poorly appreciated components of surface receptors, such as their N-glycans and their leg domains that connect ligand-binding domains to the cell surface, can have important regulatory functions.” We have toned down the description of crowding. We now have no sections with “Crowding” in the title and make clear that the effects we see may include both repulsion and crowding. The Discussion now focuses more on the important regulatory role of lower leg domains and carbohydrates, and includes both crowding and repulsion as concepts. We have included mutational studies here.

Referee #3:

This is an extremely interesting study on conformation-specific ligand binding affinities. The idea of using antibodies to trap receptors in specific conformations is powerful and should have wide applicability. I have a few minor comments that should help improve the already high quality of this paper.

1. Given the potential of conformation trapping by antibodies, the authors may want to comment on how their approach can be adapted for other receptors. Conformation-preferential nanobody could be another useful tool to trap the conformational states.

2. I'm not convinced of the "intramolecular" crowding hypothesis. If there are multiple copies of the integrin molecules in the membrane, then it's inevitable that these molecules will interact with each other, providing ample chance for "intermolecular" crowding. The physical picture of the latter scenario is simple and clear [see, e.g., Zhou, J. Phys. Chem. B 113, 7995-8005 (2009)], whereas the "intramoelcular" scenario seems hard to imagine. In any event, if the authors insist they are correct, they should add a plausible physical picture for how it works.

3. p. 5, line 3, "affinity" -- I assume the authors mean affinity for the RGD ligand. If so, this should be explicitly stated. On first reading, I thought the authors were referring to affinities for the antibodies.

We have incorporated all these suggestions, as described in the response to Reviewer 2. The reviewer should note that if crowding were between molecules in the membrane, it would favor more compact conformations, while we find the opposite- more extended molecules are favored. Our first section in the Discussion is now titled “**Methodologies for measuring intrinsic affinities and conformational equilibria of cell surface receptors.**” We have rewritten the section on crowding, and now entitle it “**Measurements of equilibria provide insights into integrin structure and function orthogonal but complementary to insights from structural biology.**”

2nd Editorial Decision

07 December 2016

Thank you for submitting a revised version of your manuscript. The manuscript has now been seen by all referees, who find that all their concerns have now been sufficiently addressed. However, before we can officially accept the manuscript there are a few editorial issues concerning the text and figures that I need you to address in a final revision.

Thank you again for giving us the chance to consider your manuscript for The EMBO Journal. I am looking forward to your final revision.

REFeree REPORTS

Referee #1:

Fine with the responses

Referee #2:

The authors have made relatively small revisions to their manuscript, but because my suggestions were minor anyway, I am satisfied and recommend acceptance. Due to the revisions, the manuscript is now easier to access, but still very heavy reading. The results however are highly relevant and interesting, and therefore I recommend acceptance. I also want to note that the authors could have made the work of the referees easier by marking up their changes. Finally I note that they misunderstood my comments on the ideal gas. If one teaches statistical mechanics, then one knows that $\Delta G = -R T \ln K$ (compare Fig. 1) follows from the chemical potential of an ideal gas (as does the law of mass action). I also mention that the crowding between the legs described here is very similar to steric repulsion between sterically stabilized colloids, a subject well studied in soft matter physics (but not in biophysics).

Referee #3:

This revision addressed my comments and is suitable for publication.

2nd Revision - authors' response

12 December 2016

Authors made the requested editorial changes.

Corresponding Author Name: Timothy A. Springer

Manuscript Number: EMBOJ-2016-95803